# Estimating High Order Gradients of the Data Distribution by Denoising

**Chenlin Meng**
Stanford University
chenlin@cs.stanford.edu

**Yang Song**
Stanford University
yangsong@cs.stanford.edu

**Wenzhe Li**
Tsinghua University
lwz21@mails.tsinghua.edu.cn

**Stefano Ermon**
Stanford University
ermon@cs.stanford.edu

## Abstract

The first order derivative of a data density can be estimated efficiently by denoising score matching, and has become an important component in many applications, such as image generation and audio synthesis. Higher order derivatives provide additional local information about the data distribution and enable new applications. Although they can be estimated via automatic differentiation of a learned density model, this can amplify estimation errors and is expensive in high dimensional settings. To overcome these limitations, we propose a method to directly estimate high order derivatives (scores) of a data density from samples. We first show that denoising score matching can be interpreted as a particular case of Tweedie's formula. By leveraging Tweedie's formula on higher order moments, we generalize denoising score matching to estimate higher order derivatives. We demonstrate empirically that models trained with the proposed method can approximate second order derivatives more efficiently and accurately than via automatic differentiation. We show that our models can be used to quantify uncertainty in denoising and to improve the mixing speed of Langevin dynamics via Ozaki discretization for sampling synthetic data and natural images.

## 1 Introduction

The first order derivative of the log data density function, also known as *score*, has found many applications including image generation [23, 24, 6], image denoising [20, 19] and audio synthesis [9]. Denoising score matching (DSM) [29] provides an efficient way to estimate the *score* of the data density from samples and has been widely used for training score-based generative models [23, 24] and denoising [20, 19]. High order derivatives of the data density, which we refer to as *high order scores*, provide a more accurate local approximation of the data density (*e.g.*, its curvature) and enable new applications. For instance, high order scores can improve the mixing speed for certain sampling methods [2, 18, 12], similar to how high order derivatives accelerate gradient descent in optimization [11]. In denoising problems, given a noisy datapoint, high order scores can be used to compute high order moments of the underlying noise-free datapoint, thus providing a way to quantify the uncertainty in denoising.

Existing methods for score estimation [8, 29, 25, 32], such as denoising score matching [29], focus on estimating the *first order* score (*i.e.*, the Jacobian of the log density). In principle, high order scores can be estimated from a learned first order score model (or even a density model) via automatic differentiation. However, this approach is computationally expensive for high dimensional data and score models parameterized by deep neural networks. For example, given a $D$ dimensional

35th Conference on Neural Information Processing Systems (NeurIPS 2021).

distribution, computing the $(n + 1)$-th order score value from an existing $n$-th order score model by automatic differentiation is on the order of $D$ times more expensive than evaluating the latter [25]. Moreover, computing higher-order scores by automatic differentiation might suffer from large estimation error, since a small training loss for the first order score does not always lead to a small estimation error for high order scores.

To overcome these limitations, we propose a new approach which directly models and estimates high order scores of a data density from samples. We draw inspiration from Tweedie's formula [4, 16], which connects the score function to a denoising problem, and show that denoising score matching (DSM) with Gaussian noise perturbation can be derived from Tweedie's formula with the knowledge of least squares regression. We then provide a generalized version of Tweedie's formula which allows us to further extend denoising score matching to estimate high order scores. In addition, we provide variance reduction techniques to improve the optimization of these newly introduced high order score estimation objectives. With our approach, we can directly parameterize high order scores and learn them efficiently, sidestepping expensive automatic differentiation.

While our theory and estimation method is applicable to scores of any order, we focus on the *second order score* (*i.e.*, the Hessian of the log density) for empirical evaluation. Our experiments show that models learned with the proposed objective can approximate second order scores more accurately than applying automatic differentiation to lower order score models. Our approach is also more computationally efficient for high dimensional data, achieving up to $500\times$ speedups for second order score estimation on MNIST. In denoising problems, there could be multiple clean datapoints consistent with a noisy observation, and it is often desirable to measure the uncertainty of denoising results. As second order scores are closely related to the covaraince matrix of the noise-free data conditioned on the noisy observation, we show that our estimated second order scores can provide extra insights into the solution of denoising problems by capturing and quantifying the uncertainty of denoising. We further show that our model can be used to improve the mixing speed of Langevin dynamics for sampling synthetic data and natural images. Our empirical results on second order scores, a special case of the general approach, demonstrate the potential and applications of our method for estimating high order scores.

## 2 Background

### 2.1 Scores of a distribution

**Definition 1.** *Given a probability density $p(\mathbf{x})$ over $\mathbb{R}^D$, we define the $k$-th order score $\mathbf{s}_k(\mathbf{x})$ : $\mathbb{R}^D \to \otimes^k \mathbb{R}^D$, where $\otimes^k$ denotes $k$-fold tensor multiplications, to be a tensor with the $(i_1, i_2, \ldots, i_k)$- th index given by $\left[\mathbf{s}_k(\mathbf{x})\right]_{i_1 i_2 \ldots i_k} \triangleq \frac{\partial^k}{\partial x_{i_1} \partial x_{i_2} \cdots \partial x_{i_k}} \log p(\mathbf{x})$, where $(i_1, i_2, \ldots, i_k) \in \{1, \cdots, D\}^k$.*

As an example, when $k = 1$, the *first order score* is the gradient of $\log p(\mathbf{x})$ w.r.t. to $\mathbf{x}$, defined as $\mathbf{s}_1(\mathbf{x}) \triangleq \nabla_\mathbf{x} \log p(\mathbf{x})$. Intuitively, this is a vector field of the steepest ascent directions for the log-density. Note that the definition of *first order score* matches the definition of (Stein) *score* [8]. When $k = 2$, the *second order score* is the Hessian of $\log p(\mathbf{x})$ w.r.t. to $\mathbf{x}$. It gives the curvature of a density function, and with $\mathbf{s}_1(\mathbf{x})$ it can provide a better local approximation to $\log p(\mathbf{x})$.

### 2.2 Denoising score matching

Given a data distribution $p_{\text{data}}(\mathbf{x})$ and a model distribution $p(\mathbf{x}; \boldsymbol{\theta})$, the *score* functions of $p_{\text{data}}(\mathbf{x})$ and $p(\mathbf{x}; \boldsymbol{\theta})$ are defined as $\mathbf{s}_1(\mathbf{x}) \triangleq \nabla_\mathbf{x} \log p_{\text{data}}(\mathbf{x})$ and $\mathbf{s}_1(\mathbf{x}; \boldsymbol{\theta}) \triangleq \nabla_\mathbf{x} \log p(\mathbf{x}; \boldsymbol{\theta})$ respectively. Denoising score matching (DSM) [29] perturbs a data sample $\mathbf{x} \sim p_{\text{data}}(\mathbf{x})$ with a pre-specified noise distribution $q_\sigma(\tilde{\mathbf{x}} \mid \mathbf{x})$ and then estimates the *score* of the perturbed data distribution $q_\sigma(\tilde{\mathbf{x}}) = \int q_\sigma(\tilde{\mathbf{x}} \mid \mathbf{x}) p_{\text{data}}(\mathbf{x}) d\mathbf{x}$ which we denote $\tilde{\mathbf{s}}_1(\tilde{\mathbf{x}}) \triangleq \nabla_{\tilde{\mathbf{x}}} \log q_\sigma(\tilde{\mathbf{x}})$. DSM uses the following objective

$$\frac{1}{2} \mathbb{E}_{p_{\text{data}}(\mathbf{x})} \mathbb{E}_{q_\sigma(\tilde{\mathbf{x}}|\mathbf{x})} \left[ \|\tilde{\mathbf{s}}_1(\tilde{\mathbf{x}}; \boldsymbol{\theta}) - \nabla_{\tilde{\mathbf{x}}} \log q_\sigma(\tilde{\mathbf{x}} \mid \mathbf{x})\|_2^2 \right]. \tag{1}$$

It is shown that under certain regularity conditions, minimizing Eq. (1) is equivalent to minimizing the score matching [8] loss between $\tilde{\mathbf{s}}_1(\tilde{\mathbf{x}}; \boldsymbol{\theta})$ and $\tilde{\mathbf{s}}_1(\tilde{\mathbf{x}})$ [29] defined as

$$\frac{1}{2} \mathbb{E}_{p_{\text{data}}(\mathbf{x})} \mathbb{E}_{q_\sigma(\tilde{\mathbf{x}}|\mathbf{x})} \left[ \|\tilde{\mathbf{s}}_1(\tilde{\mathbf{x}}; \boldsymbol{\theta}) - \tilde{\mathbf{s}}_1(\tilde{\mathbf{x}})\|_2^2 \right]. \tag{2}$$

When $q_\sigma(\tilde{\mathbf{x}} \mid \mathbf{x}) = \mathcal{N}(\tilde{\mathbf{x}}|\mathbf{x}, \sigma^2 I)$, the objective becomes

$$\mathcal{L}_{\text{DSM}}(\boldsymbol{\theta}) = \frac{1}{2}\mathbb{E}_{p_{\text{data}}(\mathbf{x})}\mathbb{E}_{q_\sigma(\tilde{\mathbf{x}}|\mathbf{x})}\left[\left\|\tilde{\mathbf{s}}_1(\tilde{\mathbf{x}};\boldsymbol{\theta}) + \frac{1}{\sigma^2}(\tilde{\mathbf{x}} - \mathbf{x})\right\|_2^2\right]. \tag{3}$$

Optimizing Eq. (3) can, intuitively, be understood as predicting $\frac{\tilde{\mathbf{x}}-\mathbf{x}}{\sigma^2}$, the added "noise" up to a constant, given the noisy input $\tilde{\mathbf{x}}$, and is thus related to denoising. Estimating the score of the noise perturbed distribution $q_\sigma(\tilde{\mathbf{x}})$ instead of the original (clean) data distribution $p_{\text{data}}(\mathbf{x})$ allows DSM to approximate scores more efficiently than other methods [8, 25]. When $\sigma$ is close to zero, $q_\sigma(\tilde{\mathbf{x}}) \approx p_{\text{data}}(\mathbf{x})$ so the score of $q_\sigma(\tilde{\mathbf{x}})$ estimated by DSM will be close to that of $p_{\text{data}}(\mathbf{x})$. When $\sigma$ is large, the estimated score for $q_\sigma(\tilde{\mathbf{x}})$ plays a crucial role in denoising [20] and learning score-based generative models [23, 24].

### 2.3 Tweedie's formula

Given a prior density $p_{\text{data}}(\mathbf{x})$, a noise distribution $q_\sigma(\tilde{\mathbf{x}}|\mathbf{x}) = \mathcal{N}(\tilde{\mathbf{x}}|\mathbf{x}, \sigma^2 I)$, and the noisy density $q_\sigma(\tilde{\mathbf{x}}) = \int p_{\text{data}}(\mathbf{x})q_\sigma(\tilde{\mathbf{x}}|\mathbf{x})d\mathbf{x}$, Tweedie's formula [16, 4] provides a close-form expression for the posterior expectation (the first moment) of $\mathbf{x}$ conditioned on $\tilde{\mathbf{x}}$:

$$\mathbb{E}[\mathbf{x} \mid \tilde{\mathbf{x}}] = \tilde{\mathbf{x}} + \sigma^2\tilde{\mathbf{s}}_1(\tilde{\mathbf{x}}), \tag{4}$$

where $\tilde{\mathbf{s}}_1(\tilde{\mathbf{x}}) \triangleq \nabla_{\tilde{\mathbf{x}}} \log q_\sigma(\tilde{\mathbf{x}})$. Equation 4 implies that given a "noisy" observation $\tilde{\mathbf{x}} \sim q_\sigma(\tilde{\mathbf{x}})$, one can compute the expectation of the "clean" datapoint $\mathbf{x}$ that may have produced $\tilde{\mathbf{x}}$. As a result, Equation 4 has become an important tool for denoising [19, 20]. We provide the proof in Appendix B.

A less widely known fact is that Tweedies' formula can be generalized to provide higher order moments of $\mathbf{x}$ given $\tilde{\mathbf{x}}$, which we will leverage to derive the objective for learning higher order scores.

## 3 Estimating Higher Order Scores by Denoising

Below we demonstrate that DSM can be derived from Tweedie's formula [4, 16]. By leveraging the generalized Tweedie's formula on high order moments of the posterior, we extend DSM to estimate higher order score functions.

### 3.1 DSM in the view of Tweedie's formula

The optimal solution to the least squares regression problem

$$\min_{\boldsymbol{\theta}} \mathbb{E}_{p_{\text{data}}(\mathbf{x})}\mathbb{E}_{q_\sigma(\tilde{\mathbf{x}}|\mathbf{x})}[\|\mathbf{h}(\tilde{\mathbf{x}};\boldsymbol{\theta}) - \mathbf{x}\|_2^2] \tag{5}$$

is well-known to be the conditional expectation $\mathbf{h}(\tilde{\mathbf{x}};\boldsymbol{\theta}^*) = \mathbb{E}[\mathbf{x} \mid \tilde{\mathbf{x}}]$. If we parameterize $\mathbf{h}(\tilde{\mathbf{x}};\boldsymbol{\theta}) = \tilde{\mathbf{x}} + \sigma^2\tilde{\mathbf{s}}_1(\tilde{\mathbf{x}};\boldsymbol{\theta})$ where $\tilde{\mathbf{s}}_1(\tilde{\mathbf{x}};\boldsymbol{\theta})$ is a first order score model with parameter $\boldsymbol{\theta}$, the least squares problem in Eq. (5) becomes equivalent to the DSM objective:

$$\min_{\boldsymbol{\theta}} \mathbb{E}_{p_{\text{data}}(\mathbf{x})}\mathbb{E}_{q_\sigma(\tilde{\mathbf{x}}|\mathbf{x})}[\|\sigma^2\tilde{\mathbf{s}}_1(\tilde{\mathbf{x}};\boldsymbol{\theta}) + \tilde{\mathbf{x}} - \mathbf{x}\|_2^2] = \min_{\boldsymbol{\theta}} 2\sigma^4 \cdot \mathcal{L}_{\text{DSM}}(\boldsymbol{\theta}). \tag{6}$$

From Tweedie's formula, we know the optimal $\boldsymbol{\theta}^*$ satisfies $\mathbf{h}(\tilde{\mathbf{x}};\boldsymbol{\theta}^*) = \tilde{\mathbf{x}} + \sigma^2\tilde{\mathbf{s}}_1(\tilde{\mathbf{x}};\boldsymbol{\theta}^*) = \mathbb{E}[\mathbf{x} \mid \tilde{\mathbf{x}}] = \tilde{\mathbf{x}}+\sigma^2\tilde{\mathbf{s}}_1(\tilde{\mathbf{x}})$, from which we can conclude that $\tilde{\mathbf{s}}_1(\tilde{\mathbf{x}};\boldsymbol{\theta}^*) = \tilde{\mathbf{s}}_1(\tilde{\mathbf{x}})$. This proves that minimizing the DSM objective in Eq. (6) recovers the first order score.

There are other ways to derive DSM. For example, [15] provides a proof based on Bayesian least squares without relying on Tweedie's formula. Stein's Unbiased Risk Estimator (SURE) [27] can also provide an alternative proof based on integration by parts. Compared to these methods, our derivation can be easily extended to learn high order scores, leveraging a more general version of Tweedie's formula.

### 3.2 Second order denoising score matching

As a warm-up, we first consider the second order score, and later generalize to any desired order. Leveraging Tweedie's formula on $\mathbb{E}[\mathbf{x}\mathbf{x}^\mathsf{T} \mid \tilde{\mathbf{x}}]$ and $\mathbb{E}[\mathbf{x} \mid \tilde{\mathbf{x}}]$, we obtain the following theorem.

**Theorem 1.** *Given a D-dimensional distribution $p(\mathbf{x})$ and $q_\sigma(\tilde{\mathbf{x}}) \triangleq \int p(\mathbf{x})q_\sigma(\tilde{\mathbf{x}}|\mathbf{x})d\mathbf{x}$, we have*

$$\mathbb{E}[\mathbf{x}\mathbf{x}^\mathsf{T} \mid \tilde{\mathbf{x}}] = \mathbf{f}(\tilde{\mathbf{x}}, \tilde{\mathbf{s}}_1, \tilde{\mathbf{s}}_2) \tag{7}$$

$$\mathbb{E}[\mathbf{x}\mathbf{x}^\mathsf{T} - \mathbf{x}\tilde{\mathbf{x}}^\mathsf{T} - \tilde{\mathbf{x}}\mathbf{x}^\mathsf{T} \mid \tilde{\mathbf{x}}] = \mathbf{h}(\tilde{\mathbf{x}}, \tilde{\mathbf{s}}_1, \tilde{\mathbf{s}}_2), \tag{8}$$

*where $\mathbf{f}(\tilde{\mathbf{x}}, \tilde{\mathbf{s}}_1, \tilde{\mathbf{s}}_2)$ and $\mathbf{h}(\tilde{\mathbf{x}}, \tilde{\mathbf{s}}_1, \tilde{\mathbf{s}}_2)$ are polynomials of $\tilde{\mathbf{x}}, \tilde{\mathbf{s}}_1(\tilde{\mathbf{x}}), \tilde{\mathbf{s}}_2(\tilde{\mathbf{x}})$ defined as*

$$\mathbf{f}(\tilde{\mathbf{x}}, \tilde{\mathbf{s}}_1, \tilde{\mathbf{s}}_2) = \tilde{\mathbf{x}}\tilde{\mathbf{x}}^\mathsf{T} + \sigma^2\tilde{\mathbf{x}}\tilde{\mathbf{s}}_1(\tilde{\mathbf{x}})^\mathsf{T} + \sigma^2\tilde{\mathbf{s}}_1(\tilde{\mathbf{x}})\tilde{\mathbf{x}}^\mathsf{T} + \sigma^4\tilde{\mathbf{s}}_2(\tilde{\mathbf{x}}) + \sigma^4\tilde{\mathbf{s}}_1(\tilde{\mathbf{x}})\tilde{\mathbf{s}}_1(\tilde{\mathbf{x}})^\mathsf{T} + \sigma^2 I, \tag{9}$$

$$\mathbf{h}(\tilde{\mathbf{x}}, \tilde{\mathbf{s}}_1, \tilde{\mathbf{s}}_2) = -\tilde{\mathbf{x}}\tilde{\mathbf{x}}^\mathsf{T} + \sigma^4\tilde{\mathbf{s}}_2(\tilde{\mathbf{x}}) + \sigma^4\tilde{\mathbf{s}}_1(\tilde{\mathbf{x}})\tilde{\mathbf{s}}_1(\tilde{\mathbf{x}})^\mathsf{T} + \sigma^2 I. \tag{10}$$

*Here $\tilde{\mathbf{s}}_1(\tilde{\mathbf{x}})$ and $\tilde{\mathbf{s}}_2(\tilde{\mathbf{x}})$ denote the first and second order scores of $q_\sigma(\tilde{\mathbf{x}})$.*

In Theorem 1, Eq. (9) is directly given by Tweedie's formula on $\mathbb{E}[\mathbf{x}\mathbf{x}^\mathsf{T} \mid \tilde{\mathbf{x}}]$, and Eq. (10) is derived from Tweedie's formula on both $\mathbb{E}[\mathbf{x} \mid \tilde{\mathbf{x}}]$ and $\mathbb{E}[\mathbf{x}\mathbf{x}^\mathsf{T} \mid \tilde{\mathbf{x}}]$. Given a noisy sample $\tilde{\mathbf{x}}$, Theorem 1 relates the second order moment of $\mathbf{x}$ to the first order score $\tilde{\mathbf{s}}_1(\tilde{\mathbf{x}})$ and second order score $\tilde{\mathbf{s}}_2(\tilde{\mathbf{x}})$ of $q_\sigma(\tilde{\mathbf{x}})$. A detailed proof of Theorem 1 is given in Appendix B.

In the same way as how we derive DSM from Tweedie's formula in Section 3.1, we can obtain higher order score matching objectives with Eq. (9) and Eq. (10) as a least squares problem.

**Theorem 2.** *Suppose the first order score $\tilde{\mathbf{s}}_1(\tilde{\mathbf{x}})$ is given, we can learn a second order score model $\tilde{\mathbf{s}}_2(\tilde{\mathbf{x}}; \boldsymbol{\theta})$ by optimizing the following objectives*

$$\boldsymbol{\theta}^* = \arg\min_{\boldsymbol{\theta}} \mathbb{E}_{p_{data}(\mathbf{x})}\mathbb{E}_{q_\sigma(\tilde{\mathbf{x}}|\mathbf{x})}\left[\left\|\mathbf{x}\mathbf{x}^\mathsf{T} - \mathbf{f}(\tilde{\mathbf{x}}, \tilde{\mathbf{s}}_1(\tilde{\mathbf{x}}), \tilde{\mathbf{s}}_2(\tilde{\mathbf{x}}; \boldsymbol{\theta}))\right\|_2^2\right], \tag{11}$$

$$\boldsymbol{\theta}^* = \arg\min_{\boldsymbol{\theta}} \mathbb{E}_{p_{data}(\mathbf{x})}\mathbb{E}_{q_\sigma(\tilde{\mathbf{x}}|\mathbf{x})}\left[\left\|\mathbf{x}\mathbf{x}^\mathsf{T} - \mathbf{x}\tilde{\mathbf{x}}^\mathsf{T} - \tilde{\mathbf{x}}\mathbf{x}^\mathsf{T} - \mathbf{h}(\tilde{\mathbf{x}}, \tilde{\mathbf{s}}_1(\tilde{\mathbf{x}}), \tilde{\mathbf{s}}_2(\tilde{\mathbf{x}}; \boldsymbol{\theta}))\right\|_2^2\right] \tag{12}$$

*where $\mathbf{f}(\cdot)$ and $\mathbf{h}(\cdot)$ are polynomials defined in Eq. (9) and Eq. (10). Assuming the model has an infinite capacity, then the optimal parameter $\boldsymbol{\theta}^*$ satisfies $\tilde{\mathbf{s}}_2(\tilde{\mathbf{x}}; \boldsymbol{\theta}^*) = \tilde{\mathbf{s}}_2(\tilde{\mathbf{x}})$ for almost any $\tilde{\mathbf{x}}$.*

Here Eq. (11) and Eq. (12) correspond to the least squares objective of Eq. (7) and Eq. (8) respectively, and have the same set of solutions assuming sufficient model capacity. In practice, we find that Eq. (12) has a much simpler form than Eq. (11), and will therefore use Eq. (12) in our experiments.

### 3.3 High order denoising score matching

Below we generalize our approach to even higher order scores by (i) leveraging Tweedie's formula to connect higher order moments of $\mathbf{x}$ conditioned on $\tilde{\mathbf{x}}$ to higher order scores of $q_\sigma(\tilde{\mathbf{x}})$; and (ii) finding the corresponding least squares objective.

**Theorem 3.** $\mathbb{E}[\otimes^n\mathbf{x}|\tilde{\mathbf{x}}] = \mathbf{f}_n(\tilde{\mathbf{x}}, \tilde{\mathbf{s}}_1, ..., \tilde{\mathbf{s}}_n)$, *where $\otimes^n\mathbf{x} \in \mathbb{R}^{D^n}$ denotes $n$-fold tensor multiplications, $\mathbf{f}_n(\tilde{\mathbf{x}}, \tilde{\mathbf{s}}_1, ..., \tilde{\mathbf{s}}_n)$ is a polynomial of $\{\tilde{\mathbf{x}}, \tilde{\mathbf{s}}_1(\tilde{\mathbf{x}}), ..., \tilde{\mathbf{s}}_n(\tilde{\mathbf{x}})\}$ and $\tilde{\mathbf{s}}_k(\tilde{\mathbf{x}})$ represents the $k$-th order score of $q_\sigma(\tilde{\mathbf{x}}) = \int p_{data}(\mathbf{x})q_\sigma(\tilde{\mathbf{x}}|\mathbf{x})d\mathbf{x}$.*

Theorem 3 shows that there exists an equality between (high order) moments of the posterior distribution of $\mathbf{x}$ given $\tilde{\mathbf{x}}$ and (high order) scores with respect to $\tilde{\mathbf{x}}$. To get some intuition, for $n = 2$ the polynomial $\mathbf{f}_2(\tilde{\mathbf{x}}, \tilde{\mathbf{s}}_1, \tilde{\mathbf{s}}_2)$ is simply the function $\mathbf{f}$ in Eq. (9). In Appendix B, we provide a recursive formula for obtaining the coefficients of $\mathbf{f}_n$ in closed form.

Leveraging Theorem 3 and the least squares estimation of $\mathbb{E}[\otimes^k\mathbf{x}|\tilde{\mathbf{x}}]$, we can construct objectives for approximating the $k$-th order scores $\tilde{\mathbf{s}}_k(\tilde{\mathbf{x}})$ as in the following theorem.

**Theorem 4.** *Given score functions $\tilde{\mathbf{s}}_1(\tilde{\mathbf{x}}), ..., \tilde{\mathbf{s}}_{k-1}(\tilde{\mathbf{x}})$, a $k$-th order score model $\tilde{\mathbf{s}}_k(\tilde{\mathbf{x}}; \boldsymbol{\theta})$, and*

$$\boldsymbol{\theta}^* = \arg\min_{\boldsymbol{\theta}} \mathbb{E}_{p_{data}(\mathbf{x})}\mathbb{E}_{q_\sigma(\tilde{\mathbf{x}}|\mathbf{x})}[\|\otimes^k \mathbf{x} - \mathbf{f}_k(\tilde{\mathbf{x}}, \tilde{\mathbf{s}}_1(\tilde{\mathbf{x}}), ..., \tilde{\mathbf{s}}_{k-1}(\tilde{\mathbf{x}}), \tilde{\mathbf{s}}_k(\tilde{\mathbf{x}}; \boldsymbol{\theta}))\|^2].$$

*We have $\tilde{\mathbf{s}}_k(\tilde{\mathbf{x}}; \boldsymbol{\theta}^*) = \tilde{\mathbf{s}}_k(\tilde{\mathbf{x}})$ for almost all $\tilde{\mathbf{x}}$.*

As previously discussed, when $\sigma$ approaches 0 such that $q_\sigma(\tilde{\mathbf{x}}) \approx p_{data}(\mathbf{x})$, $\tilde{\mathbf{s}}_k(\tilde{\mathbf{x}}; \boldsymbol{\theta}^*)$ well-approximates the $k$-th order score of $p_{data}(\mathbf{x})$.

# 4 Learning Second Order Score Models

Although our theory can be applied to scores of any order, we focus on second order scores for empirical analysis. In this section, we discuss the parameterization and empirical performance of the learned second order score models.

## 4.1 Instantiating objectives for second order score models

In practice, we find that Eq. (12) has a much simpler expression than Eq. (11). Therefore, we propose to parameterize $\tilde{\mathbf{s}}_2(\tilde{\mathbf{x}})$ with a model $\tilde{\mathbf{s}}_2(\tilde{\mathbf{x}}; \boldsymbol{\theta})$, and optimize $\tilde{\mathbf{s}}_2(\tilde{\mathbf{x}}; \boldsymbol{\theta})$ with Eq. (12), which can be simplified to the following after combining Eq. (10) and Eq. (12):

$$\mathcal{L}_{D_2\text{SM}}(\boldsymbol{\theta}) \triangleq \mathbb{E}_{p_{\text{data}}(\mathbf{x})}\mathbb{E}_{q_\sigma(\tilde{\mathbf{x}}|\mathbf{x})}\left[\left\|\tilde{\mathbf{s}}_2(\tilde{\mathbf{x}}; \boldsymbol{\theta}) + \tilde{\mathbf{s}}_1(\tilde{\mathbf{x}}; \boldsymbol{\theta})\tilde{\mathbf{s}}_1(\tilde{\mathbf{x}}; \boldsymbol{\theta})^{\mathsf{T}} + \frac{I - \mathbf{z}\mathbf{z}^{\mathsf{T}}}{\sigma^2}\right\|_2^2\right], \quad (13)$$

where $\mathbf{z} \triangleq \frac{\tilde{\mathbf{x}} - \mathbf{x}}{\sigma}$. Note that Eq. (13) requires knowing the first order score $\tilde{\mathbf{s}}_1(\tilde{\mathbf{x}})$ in order to train the second order score model $\tilde{\mathbf{s}}_2(\tilde{\mathbf{x}}; \boldsymbol{\theta})$. We therefore use the following hybrid objective to simultaneously train both $\tilde{\mathbf{s}}_1(\tilde{\mathbf{x}}; \boldsymbol{\theta})$ and $\tilde{\mathbf{s}}_2(\tilde{\mathbf{x}}; \boldsymbol{\theta})$:

$$\mathcal{L}_{\text{joint}}(\boldsymbol{\theta}) = \mathcal{L}_{D_2\text{SM}}(\boldsymbol{\theta}) + \gamma \cdot \mathcal{L}_{\text{DSM}}(\boldsymbol{\theta}), \quad (14)$$

where $\mathcal{L}_{\text{DSM}}(\boldsymbol{\theta})$ is defined in Eq. (3) and $\gamma \in \mathbb{R}_{>0}$ is a tunable coefficient. The expectation for $\mathcal{L}_{D_2\text{SM}}(\boldsymbol{\theta})$ and $\mathcal{L}_{\text{DSM}}(\boldsymbol{\theta})$in Eq. (14) can be estimated with samples, and we optimize the following unbiased estimator

$$\hat{\mathcal{L}}_{\text{joint}}(\boldsymbol{\theta}) = \frac{1}{N}\sum_{i=1}^{N}\left[\left\|\tilde{\mathbf{s}}_2(\tilde{\mathbf{x}}_i; \boldsymbol{\theta}) + \tilde{\mathbf{s}}_1(\tilde{\mathbf{x}}_i; \boldsymbol{\theta})\tilde{\mathbf{s}}_1(\tilde{\mathbf{x}}_i; \boldsymbol{\theta})^{\mathsf{T}} + \frac{I - \mathbf{z}_i\mathbf{z}_i^{\mathsf{T}}}{\sigma^2}\right\|_2^2 + \frac{\gamma}{2}\left\|\tilde{\mathbf{s}}_1(\tilde{\mathbf{x}}_i; \boldsymbol{\theta}) + \frac{\mathbf{z}_i}{\sigma}\right\|_2^2\right], \quad (15)$$

where we define $\mathbf{z}_i \triangleq \frac{\tilde{\mathbf{x}}_i - \mathbf{x}_i}{\sigma}$, and $\{\tilde{\mathbf{x}}_i\}_{i=1}^{N}$ are samples from $q_\sigma(\tilde{\mathbf{x}}) = \int p_{\text{data}}(\mathbf{x})q_\sigma(\tilde{\mathbf{x}}|\mathbf{x})d\mathbf{x}$ which can be obtained by adding noise to samples from $p_{\text{data}}(\mathbf{x})$. Similarly to DSM, when $\sigma \to 0$, the optimal model $\tilde{\mathbf{s}}_2(\tilde{\mathbf{x}}; \boldsymbol{\theta}^*)$ that minimizes Eq. (15) will be close to the second order score of $p_{\text{data}}(\mathbf{x})$ because $q_\sigma(\tilde{\mathbf{x}}) \approx p_{\text{data}}(\mathbf{x})$. When $\sigma$ is large, the learned $\tilde{\mathbf{s}}_2(\tilde{\mathbf{x}}; \boldsymbol{\theta})$ can be applied to tasks such as uncertainty quantification for denoising, which will be discussed in Section 5.

For downstream tasks that require only the diagonal of $\tilde{\mathbf{s}}_2$, we can instead optimize a simpler objective

$$\mathcal{L}_{\text{joint-diag}}(\boldsymbol{\theta}) \triangleq \mathcal{L}_{D_2\text{SM-diag}}(\boldsymbol{\theta}) + \gamma \cdot \mathcal{L}_{\text{DSM}}(\boldsymbol{\theta}), \quad \text{where} \quad (16)$$

$$\mathcal{L}_{D_2\text{SM-diag}}(\boldsymbol{\theta}) \triangleq \mathbb{E}_{p_{\text{data}}(\mathbf{x})}\mathbb{E}_{q_\sigma(\tilde{\mathbf{x}}|\mathbf{x})}\left[\left\|\text{diag}(\tilde{\mathbf{s}}_2(\tilde{\mathbf{x}}; \boldsymbol{\theta})) + \tilde{\mathbf{s}}_1(\tilde{\mathbf{x}}; \boldsymbol{\theta}) \odot \tilde{\mathbf{s}}_1(\tilde{\mathbf{x}}; \boldsymbol{\theta}) + \frac{\mathbf{1} - \mathbf{z} \odot \mathbf{z}}{\sigma^2}\right\|_2^2\right]. \quad (17)$$

Here diag$(\cdot)$ denotes the diagonal of a matrix and $\odot$ denotes element-wise multiplication. Optimizing Eq. (16) only requires parameterizing diag$(\tilde{\mathbf{s}}_2(\tilde{\mathbf{x}}; \boldsymbol{\theta}))$, which can significantly reduce the memory and computational cost for training and running the second order score model. Similar to $\hat{\mathcal{L}}_{\text{joint}}(\boldsymbol{\theta})$, we estimate the expectation in Eq. (17) with empirical means.

## 4.2 Parameterizing second order score models

In practice, the performance of learning second order scores is affected by model parameterization. As many real world data distributions (*e.g.*, images) tend to lie on low dimensional manifolds [13, 3, 21], we propose to parametrize $\tilde{\mathbf{s}}_2(\tilde{\mathbf{x}}; \boldsymbol{\theta})$ with low rank matrices defined as below

$$\tilde{\mathbf{s}}_2(\tilde{\mathbf{x}}; \boldsymbol{\theta}) = \boldsymbol{\alpha}(\tilde{\mathbf{x}}; \boldsymbol{\theta}) + \boldsymbol{\beta}(\tilde{\mathbf{x}}; \boldsymbol{\theta})\boldsymbol{\beta}(\tilde{\mathbf{x}}; \boldsymbol{\theta})^{\mathsf{T}},$$

where $\boldsymbol{\alpha}(\cdot; \boldsymbol{\theta}) : \mathbb{R}^D \to \mathbb{R}^{D \times D}$ is a diagonal matrix, $\boldsymbol{\beta}(\cdot; \boldsymbol{\theta}) : \mathbb{R}^D \to \mathbb{R}^{D \times r}$ is a matrix with shape $D \times r$, and $r \leqslant D$ is a positive integer.

## 4.3 Antithetic sampling for variance reduction

As the standard deviation of the perturbed noise $\sigma$ approximates zero, training score models with denoising methods could suffer from a high variance. Inspired by a variance reduction method for DSM [30, 26], we propose a variance reduction method for $D_2$SM

$$\mathcal{L}_{D_2\text{SM-VR}} = \mathbb{E}_{\mathbf{x} \sim p_{\text{data}}(\mathbf{x})}\mathbb{E}_{\mathbf{z} \sim \mathcal{N}(0, I)}\left[\boldsymbol{\psi}(\tilde{\mathbf{x}}_+)^2 + \boldsymbol{\psi}(\tilde{\mathbf{x}}_-)^2 + 2\frac{\mathbf{I} - \mathbf{z}\mathbf{z}^{\mathsf{T}}}{\sigma^2} \odot (\boldsymbol{\psi}(\tilde{\mathbf{x}}_+) + \boldsymbol{\psi}(\tilde{\mathbf{x}}_-) - 2\boldsymbol{\psi}(\mathbf{x}))\right],$$

where $\tilde{\mathbf{x}}_+ = \mathbf{x} + \sigma\mathbf{z}$, $\tilde{\mathbf{x}}_- = \mathbf{x} - \sigma\mathbf{z}$ and $\psi = \tilde{\mathbf{s}}_2 + \tilde{\mathbf{s}}_1\tilde{\mathbf{s}}_1^\top$. Instead of using independent noise samples, we apply antithetic sampling and use two correlated (opposite) noise vectors centered at $\mathbf{x}$. Similar to Eq. (14), we define $\mathcal{L}_{\text{joint-VR}} = \mathcal{L}_{D_2\text{SM-VR}} + \gamma \cdot \mathcal{L}_{\text{DSM-VR}}$, where $\mathcal{L}_{\text{DSM-VR}}$ is proposed in [30].

We empirically study the role of variance reduction (VR) in training models with DSM and $D_2$SM. We observe that VR is crucial for both DSM and $D_2$SM when $\sigma$ is approximately zero, but is optional when $\sigma$ is large enough. To see this, we consider a 2-d Gaussian distribution $\mathcal{N}(0, I)$ and train $\tilde{\mathbf{s}}_1(\tilde{\mathbf{x}}; \boldsymbol{\theta})$ and $\tilde{\mathbf{s}}_2(\tilde{\mathbf{x}}; \boldsymbol{\theta})$ with DSM and $D_2$SM respectively. We plot the learning curves in Figs. 1a and 1b, and visualize the first dimension of the estimated scores for multiple noise scales $\sigma$ in Figs. 1c and 1d. We observe that when $\sigma = 0.001$, both DSM and $D_2$SM have trouble converging after a long period of training, while the VR counterparts converge quickly (see Fig. 1). When $\sigma$ gets larger, DSM and $D_2$SM without VR can both converge quickly and provide reasonable score estimations (Figs. 1c and 1d). We provide extra details in Appendix C.

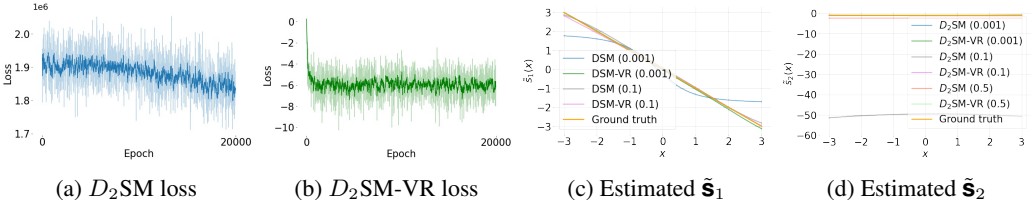

| (a) $D_2$SM loss | (b) $D_2$SM-VR loss | (c) Estimated $\tilde{\mathbf{s}}_1$ | (d) Estimated $\tilde{\mathbf{s}}_2$ |

Figure 1: From left to right: (a) $D_2$SM loss without variance reduction ($\sigma = 10^{-3}$). (b) $D_2$SM loss with variance reduction ($\sigma = 10^{-3}$). (c) Estimated $\tilde{\mathbf{s}}_1$. (d) Estimated $\tilde{\mathbf{s}}_2$, where the estimation for $D_2$SM (0.001) is too far from the ground truth to appear on the plot.

## 4.4 The accuracy and efficiency of learning second order scores

We show that the proposed method can estimate second order scores more efficiently and accurately than those obtained by automatic differentiation of a first order score model trained with DSM. We observe in our experiments that $\tilde{\mathbf{s}}_1(\tilde{\mathbf{x}}; \boldsymbol{\theta})$ jointly optimized via $\hat{\mathcal{L}}_{\text{joint}}$ or $\hat{\mathcal{L}}_{\text{joint-diag}}$ has a comparable empirical performance as trained directly by DSM, so we optimize $\tilde{\mathbf{s}}_1(\tilde{\mathbf{x}}; \boldsymbol{\theta})$ and $\tilde{\mathbf{s}}_2(\tilde{\mathbf{x}}; \boldsymbol{\theta})$ jointly in later experiments. We provide additional experimental details in Appendix C.

**Learning accuracy** We consider three synthetic datasets whose ground truth scores are available—a 100-dimensional correlated multivariate normal distribution and two high dimensional mixture of logistics distributions in Table 1. We study the performance of estimating $\tilde{\mathbf{s}}_2$ and the diagonal of $\tilde{\mathbf{s}}_2$. For the baseline, we estimate *second order scores* by taking automatic differentiation of $\tilde{\mathbf{s}}_1(\tilde{\mathbf{x}}; \boldsymbol{\theta})$ trained jointly with $\tilde{\mathbf{s}}_2(\tilde{\mathbf{x}}; \boldsymbol{\theta})$ using Eq. (15) or Eq. (17). As mentioned previously, $\tilde{\mathbf{s}}_1(\tilde{\mathbf{x}}; \boldsymbol{\theta})$ trained with the joint method has the same empirical performance as trained directly with DSM. For our method, we directly evaluate $\tilde{\mathbf{s}}_2(\tilde{\mathbf{x}}; \boldsymbol{\theta})$. We compute the mean squared error between estimated *second order scores* and the ground truth score of the *clean* data since we use small $\sigma$ and $q_\sigma(\tilde{\mathbf{x}}) \approx p_{\text{data}}(\mathbf{x})$ (see Table 1). We observe that $\tilde{\mathbf{s}}_2(\tilde{\mathbf{x}}; \boldsymbol{\theta})$ achieves better performance than the gradients of $\tilde{\mathbf{s}}_1(\tilde{\mathbf{x}}; \boldsymbol{\theta})$.

Table 1: Mean squared error between the estimated *second order scores* and the ground truth on $10^5$ test samples. Each setup is trained with three random seeds and multiple noise scales $\sigma$.

| Methods | $\sigma = 0.01$ | $\sigma = 0.05$ | $\sigma = 0.1$ | Methods | $\sigma = 0.01$ | $\sigma = 0.05$ | $\sigma = 0.1$ |
|---|---|---|---|---|---|---|---|
| Multivariate normal (100-d) | | | | Mixture of logistics diagonal estimation (50-d, 20 mixtures) | | | |
| $\tilde{\mathbf{s}}_1$ grad (DSM) | 43.80$\pm$0.012 | 43.76$\pm$0.001 | 43.75$\pm$0.001 | $\tilde{\mathbf{s}}_1$ grad (DSM-VR) | 26.41$\pm$0.55 | 26.13$\pm$0.53 | 25.39$\pm$0.50 |
| $\tilde{\mathbf{s}}_1$ grad (DSM-VR) | 9.40$\pm$0.049 | 9.39$\pm$0.015 | 9.21$\pm$0.020 | $\tilde{\mathbf{s}}_2$ (Ours) | **18.43$\pm$0.11** | **18.50$\pm$0.25** | **17.88$\pm$0.15** |
| $\tilde{\mathbf{s}}_2$ (Ours, $r = 15$) | 7.12$\pm$0.319 | 6.91$\pm$0.078 | 7.03$\pm$0.039 | Mixture of logistics diagonal estimation (80-d, 20 mixtures) | | | |
| $\tilde{\mathbf{s}}_2$ (Ours, $r = 20$) | 5.24$\pm$0.065 | 5.07$\pm$0.047 | 5.13$\pm$0.065 | $\tilde{\mathbf{s}}_1$ grad (DSM-VR) | 32.80$\pm$0.34 | 32.44$\pm$0.30 | 31.51$\pm$0.43 |
| $\tilde{\mathbf{s}}_2$ (Ours, $r = 30$) | **1.76$\pm$0.038** | **2.05$\pm$0.544** | **1.76$\pm$0.045** | $\tilde{\mathbf{s}}_2$ (Ours) | **21.68$\pm$0.18** | **22.23$\pm$0.08** | **22.18$\pm$0.08** |

**Computational efficiency** Computing the gradients of $\tilde{\mathbf{s}}_1(\tilde{\mathbf{x}}; \boldsymbol{\theta})$ via automatic differentiation can be expensive for high dimensional data and deep neural networks. To see this, we consider two models—a 3-layer MLP and a U-Net [17], which is used for image experiments in the subsequent sections. We consider a 100-d data distribution for the MLP model and a 784-d data distribution for the U-Net. We parameterize $\tilde{\mathbf{s}}_1$ and $\tilde{\mathbf{s}}_2$ with the same model architecture and use a batch size of 10 for both settings. We report the wall-clock time averaged in 7 runs used for estimating second order scores during test time on a TITAN Xp GPU in Table 2. We observe that $\tilde{\mathbf{s}}_2(\tilde{\mathbf{x}}; \boldsymbol{\theta})$ is 500$\times$ faster than using automatic differentiation for $\tilde{\mathbf{s}}_1(\tilde{\mathbf{x}}; \boldsymbol{\theta})$ on the MNIST dataset.

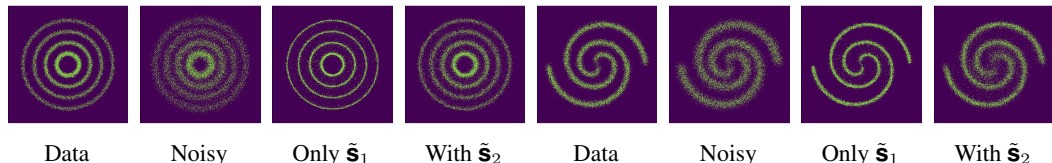

| Data | Noisy | Only $\tilde{\mathbf{s}}_1$ | With $\tilde{\mathbf{s}}_2$ | Data | Noisy | Only $\tilde{\mathbf{s}}_1$ | With $\tilde{\mathbf{s}}_2$ |

Figure 2: Denoising 2-d synthetic data. The incorporation of $\tilde{\mathbf{s}}_2$ improves uncertainty quantification.

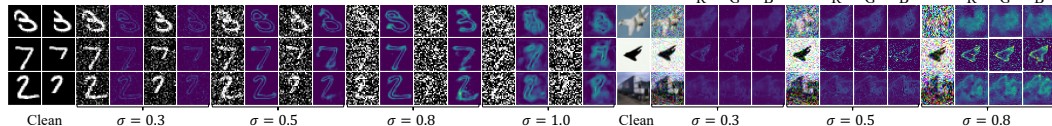

Figure 3: Visualizations of the estimated covariance matrix diagonals on MNIST and CIFAR-10. For CIFAR-10 images, we visualize the diagonal for R, G, B channels separately. Images corrupted with more noise tend to have larger covariance values, indicating larger uncertainty in denoising. Pixels in background have smaller values than pixels near edges, indicating more confident denoising.

# 5 Uncertainty Quantification with Second Order Score Models

Our second order score model $\tilde{\mathbf{s}}_2(\tilde{\mathbf{x}}; \boldsymbol{\theta})$ can capture and quantify the uncertainty of denoising on synthetic and real world image datasets, based on the following result by combining Eqs. (4) and (9)

$$\text{Cov}[\mathbf{x} \mid \tilde{\mathbf{x}}] \triangleq \mathbb{E}[\mathbf{x}\mathbf{x}^{\mathsf{T}} \mid \tilde{\mathbf{x}}] - \mathbb{E}[\mathbf{x} \mid \tilde{\mathbf{x}}]\mathbb{E}[\mathbf{x} \mid \tilde{\mathbf{x}}]^{\mathsf{T}} = \sigma^4\tilde{\mathbf{s}}_2(\tilde{\mathbf{x}}) + \sigma^2 I \qquad (18)$$

By estimating $\text{Cov}[\mathbf{x} \mid \tilde{\mathbf{x}}]$ via $\tilde{\mathbf{s}}_2(\tilde{\mathbf{x}})$, we gain insights into how pixels are correlated with each other under denoising settings, and which part of the pixels has large uncertainty. To examine the uncertainty given by our $\tilde{\mathbf{s}}_2(\tilde{\mathbf{x}}; \boldsymbol{\theta})$, we perform the following experiments (details in Appendix D).

**Synthetic experiments** We first consider 2-d synthetic datasets shown in Fig. 2, where we train $\tilde{\mathbf{s}}_1(\tilde{\mathbf{x}}; \boldsymbol{\theta})$ and $\tilde{\mathbf{s}}_2(\tilde{\mathbf{x}}; \boldsymbol{\theta})$ jointly with $\mathcal{L}_{\text{joint}}$. Given the trained score models, we estimate $\mathbb{E}[\mathbf{x} \mid \tilde{\mathbf{x}}]$ and $\text{Cov}[\mathbf{x} \mid \tilde{\mathbf{x}}]$ using Eq. (4) and Eq. (18). We approximate the posterior distribution $p(\mathbf{x}|\tilde{\mathbf{x}})$ with a conditional normal distribution $\mathcal{N}(\mathbf{x} \mid \mathbb{E}[\mathbf{x} \mid \tilde{\mathbf{x}}], \text{Cov}[\mathbf{x} \mid \tilde{\mathbf{x}}])$. We compare our result with that of Eq. (4), which only utilizes $\tilde{\mathbf{s}}_1$ (see Fig. 2). We observe that unlike Eq. (4), which is a point estimator, the incorporation of covariance matrices (estimated by $\tilde{\mathbf{s}}_2(\tilde{\mathbf{x}}; \boldsymbol{\theta})$) captures uncertainty in denoising.

**Covariance diagonal visualizations** We visualize the diagonal of the estimated $\text{Cov}[\mathbf{x} \mid \tilde{\mathbf{x}}]$ for MNIST and CIFAR-10 [10] in Fig. 3. We find that the diagonal values are in general larger for pixels near the edges where there are multiple possibilities corresponding to the same noisy pixel. The diagonal values are smaller for the background pixels where there is less uncertainty. We also observe that covariance matrices corresponding to smaller noise scales tend to have smaller values on the diagonals, implying that the more noise an image has, the more uncertain the denoised results are.

**Full convariance visualizations** We visualize the eigenvectors (sorted by eigenvalues) of $\text{Cov}[\mathbf{x} \mid \tilde{\mathbf{x}}]$ estimated by $\tilde{\mathbf{s}}_2(\tilde{\mathbf{x}}; \boldsymbol{\theta})$ in Fig. 4. We observe that they can correspond to different digit identities, indicating uncertainty in the identity of the denoised image. This suggests $\text{Cov}[\mathbf{x} \mid \tilde{\mathbf{x}}]$ can capture additional information for uncertainty beyond its diagonal.

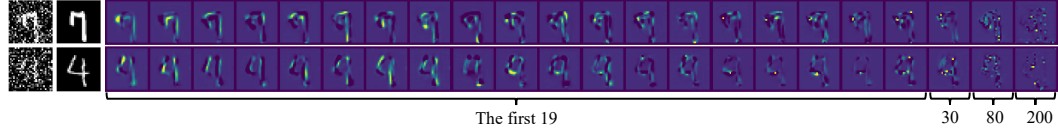

Figure 4: Eigenvectors of the estimated covariance matrix on MNIST. The first column shows the noisy images ($\sigma = 0.5$) and the second column shows clean images. The remaining columns show the first 19, plus the 30, 80 and 200-th eigenvectors of the matrix. We can see digit 7 and 9 in the eigenvectors corresponding to the noisy 7, and digit 4 and 9 in the second row, which implies that the estimated covariance matrix can capture different possibilities of the denoising results.

# 6 Sampling with Second Order Score Models

Here we show that our second order score model $\tilde{\mathbf{s}}_2(\tilde{\mathbf{x}}; \boldsymbol{\theta})$ can be used to improve the mixing speed of Langevin dynamics sampling.

Table 2: Speed analysis: direct modeling vs. autodiff.

| Method  Dimension | $D = 100$ (MLP) | $D = 784$ (U-Net) |
|---|---|---|
| Autodiff | $32100 \pm 156$ $\mu s$ | $34600 \pm 194$ ms |
| Ours (rank=20) | $\mathbf{380 \pm 7.9}$ $\mu s$ | $\mathbf{67.9 \pm 1.93}$ ms |
| Ours (rank=50) | $\mathbf{377 \pm 10.8}$ $\mu s$ | $72.5 \pm 1.93$ ms |
| Ours (rank=200) | $546 \pm 1.91$ $\mu s$ | $68.8 \pm 1.02$ ms |
| Ours (rank=1000) | $1840 \pm 97.1$ $\mu s$ | $69.4 \pm 2.63$ ms |

Table 3: ESS on synthetic datasets. Datasets are shown in Fig. 5. We use 32 chains, each with length 10000 and 1000 burn-in steps.

| | | Dataset 1 | Dataset 2 |
|---|---|---|---|
| Method | Metric | ESS ↑ | ESS ↑ |
| Langevin | | 21.81 | 26.33 |
| Ozaki | | **28.89** | **46.57** |

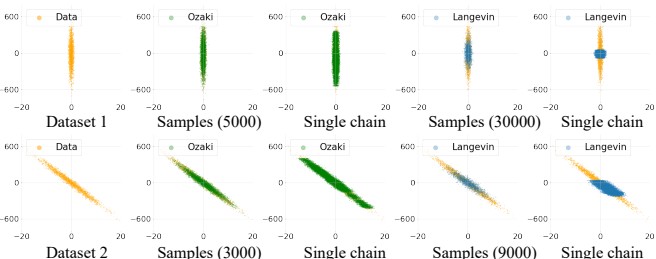

Figure 5: Sampling with Ozaki and Langevin dynamics. We tune the optimal step size separately for both methods. The number in the parenthesis (Column 2 and 4) stands for the iterations used for sampling. We observe that Ozaki obtains more reasonable samples than Langevin dynamics using 1/6 or 1/3 iterations. Column 3 and 5 show samples within a single chain with length 31000 and 1000 burn-in steps.

## 6.1 Background on the sampling methods

**Langevin dynamics** Langevin dynamics [1, 31] samples from $p_{\text{data}}(\mathbf{x})$ using the first order score function $\mathbf{s}_1(\mathbf{x})$. Given a prior distribution $\pi(\mathbf{x})$, a fixed step size $\epsilon > 0$ and an initial value $\tilde{\mathbf{x}}_0 \sim \pi(\mathbf{x})$, Langevin dynamics update the samples iteratively as follows

$$\tilde{\mathbf{x}}_t = \tilde{\mathbf{x}}_{t-1} + \frac{\epsilon}{2}\mathbf{s}_1(\tilde{\mathbf{x}}_{t-1}) + \sqrt{\epsilon}\mathbf{z}_t, \tag{19}$$

where $\mathbf{z}_t \sim \mathcal{N}(0, I)$. As $\epsilon \to 0$ and $t \to \infty$, $\tilde{\mathbf{x}}_t$ is a sample from $p_{\text{data}}(\mathbf{x})$ under suitable conditions.

**Ozaki sampling** Langevin dynamics with Ozaki discretization [28] leverages second order information in $\mathbf{s}_2(\mathbf{x})$ to pre-condition Langevin dynamics:

$$\tilde{\mathbf{x}}_t = \tilde{\mathbf{x}}_{t-1} + M_{t-1}\mathbf{s}_1(\tilde{\mathbf{x}}_{t-1}) + \Sigma_{t-1}^{1/2}\mathbf{z}_t, \ \mathbf{z}_t \sim \mathcal{N}(0, I) \tag{20}$$

where $M_{t-1} = (e^{\epsilon\mathbf{s}_2(\tilde{\mathbf{x}}_{t-1})} - I)\mathbf{s}_2(\tilde{\mathbf{x}}_{t-1})^{-1}$ and $\Sigma_{t-1} = (e^{2\epsilon\mathbf{s}_2(\tilde{\mathbf{x}}_{t-1})} - I)\mathbf{s}_2(\tilde{\mathbf{x}}_{t-1})^{-1}$. It is shown that under certain conditions, this variation can improve the convergence rate of Langevin dynamics [2]. In general, Eq. (20) is expensive to compute due to inversion, exponentiation and taking square root of matrices, so we simplify Eq. (20) by approximating $\mathbf{s}_2(\tilde{\mathbf{x}}_{t-1})$ with its diagonal in practice.

In our experiments, we only consider Ozaki sampling with $\mathbf{s}_2$ replaced by its diagonal in Eq. (20). As we use small $\sigma$, $\tilde{\mathbf{s}}_1 \approx \mathbf{s}_1$ and $\tilde{\mathbf{s}}_2 \approx \mathbf{s}_2$. We observe that $\text{diag}(\tilde{\mathbf{s}}_2(\tilde{\mathbf{x}}; \boldsymbol{\theta}))$ in Ozaki sampling can be computed in parallel with $\tilde{\mathbf{s}}_1(\tilde{\mathbf{x}}; \boldsymbol{\theta})$ on modern GPUs, making the wall-clock time per iteration of Ozaki sampling comparable to that of Langevin dynamics. Since we only use the diagonal of $\tilde{\mathbf{s}}_2(\tilde{\mathbf{x}}; \boldsymbol{\theta})$ in sampling, we can directly learn the diagonal of $\tilde{\mathbf{s}}_2(\tilde{\mathbf{x}})$ with Eq. (16).

## 6.2 Synthetic datasets

We first consider 2-d synthetic datasets in Fig. 5 to compare the mixing speed of Ozaki sampling with Langevin dynamics. We search the optimal step size for each method and observe that Ozaki sampling can use a larger step size and converge faster than Langevin dynamics (see Fig. 5). We use the optimal step size for both methods and report the smallest effective sample size (ESS) of all the dimensions [22, 5] in Table 3. We observe that Ozaki sampling has better ESS values than Langevin dynamics, implying faster mixing speed. Even when using the same step size, Ozaki sampling still converges faster than Langevin dynamics on the two-model Gaussian dataset we consider (see Fig. 6). In all the experiments, we use $\sigma = 0.1$ and we provide more experimental details in Appendix E.

## 6.3 Image datasets

Ozaki discretization with learned $\tilde{\mathbf{s}}_2(\tilde{\mathbf{x}}; \boldsymbol{\theta})$ produces more diverse samples and improve the mixing speed of Langevin dynamics on image datasets (see Fig. 7) To see this, we select ten different digits from MNIST test set and initialize 1000 different sampling chains for each image. We update the

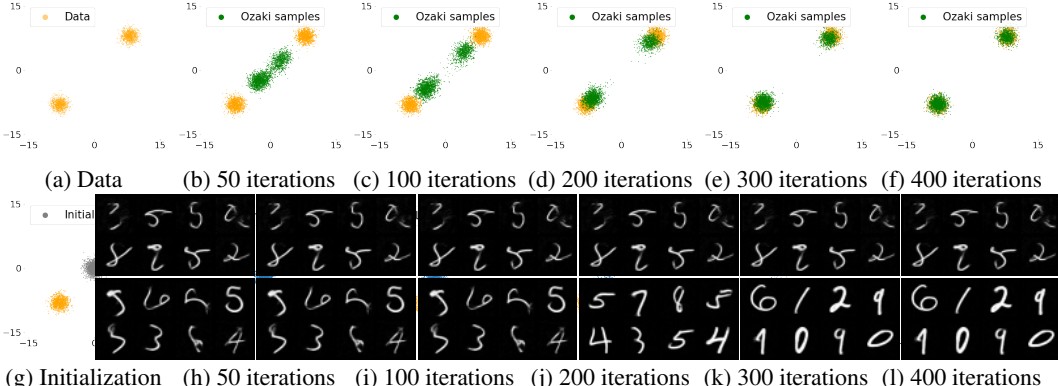

(a) Data    (b) 50 iterations    (c) 100 iterations    (d) 200 iterations    (e) 300 iterations    (f) 400 iterations

(g) Initialization    (h) 50 iterations    (i) 100 iterations    (j) 200 iterations    (k) 300 iterations    (l) 400 iterations

Figure 6: Sampling a two mode distribution. We use the same step size $\epsilon = 0.01$ for both methods. We observe that Ozaki sampling converges faster than Langevin sampling.

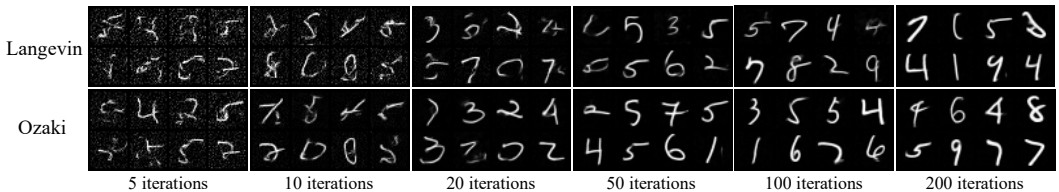

Figure 7: Sampling on MNIST. We observe that Ozaki sampling converges faster than Langevin dynamics. We use step size $\sigma = 0.02$ and initialize the chain with Gaussian noise for both methods.

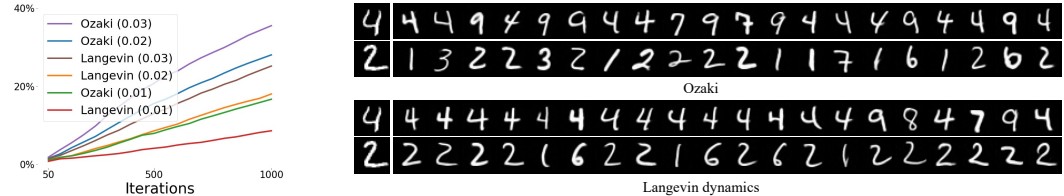

(a) Percentage of changes in class label w.r.t. iterations.

(b) Different chains initialized with the same left panel image after 1000 iterations of update with step size $\epsilon = 0.03$.

Figure 8: Sample diversity analysis. The number in the parenthesis in Fig. 8a denotes the step size. We initialize the chain with MNIST test images and report the percentage of images that have changed class labels from the initialization w.r.t. sampling iterations. We observe that Ozaki sampling has more diverse samples.

chains with Ozaki sampling and report the percentage of images that have class label changes after a fixed number of sampling iterations in Fig. 8a. We compare the results with Langevin dynamics with the same setting and observe that Ozaki sampling has more diverse samples within the same chain in a fixed amount of iterations. We provide more details in Appendix E.

# 7 Conclusion

We propose a method to directly estimate *high order scores* of a data density from samples. We first study the connection between Tweedie's formula and denoising score matching (DSM) through the lens of least squares regression. We then leverage Tweedie's formula on higher order moments, which allows us to generalize denoising score matching to estimate scores of any desired order. We demonstrate empirically that models trained with the proposed method can approximate second order scores more efficiently and accurately than applying automatic differentiation to a learned first order score model. In addition, we show that our models can be used to quantify uncertainty in denoising and to improve the mixing speed of Langevin dynamics via Ozaki discretization for sampling synthetic data and natural images. Besides the applications studied in this paper, it would be interesting to study the application of high order scores for out of distribution detection. Due to limited

computational resources, we only consider low resolution image datasets in this work. However, as a direct next step, we can apply our method to higher-resolution image datasets and explore its application to improve the sampling speed of score-based models [23, 24, 6] with Ozaki sampling. In general, when approximating the high-order scores with a diagonal or a low rank matrix, our training cost is comparable to standard denoising score matching, which is scalable to higher dimensional data. A larger rank typically requires more computation but could give better approximations to second-order scores. While we focused on images, this approach is likely applicable to other data modalities such as speech.

## Acknowledgements

The authors would like to thank Jiaming Song and Lantao Yu for constructive feedback. This research was supported by NSF (#1651565, #1522054, #1733686), ONR (N000141912145), AFOSR (FA95501910024), ARO (W911NF-21-1-0125) and Sloan Fellowship.

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
