# A Related Work

Existing methods for score estimation focus mainly on estimating the first order score of the data distribution. For instance, score matching [8] approximates the first order score by minimizing the Fisher divergence between the data distribution and model distribution. Sliced score matching [25] and finite-difference score matching [14] provide alternatives to estimating the first order score by approximating the score matching loss [8] using Hutchinson's trace estimator [7] and finite difference respectively. Denoising score matching (DSM) [29] estimates the first order score of a noise perturbed data distribution by predicting the added perturbed "noise" given a noisy observation. However, none of these methods can directly model and estimate higher order scores. In this paper we study DSM from the perspective of Tweedie's formula and propose a method for estimating high order scores. There are also other ways to derive DSM without using Tweedie's formula. For example, [15] provides a proof based on Bayesian least squares estimation. Stein's Unbiased Risk Estimator (SURE) [27] can also provide an alternative proof based on integration by parts. In contrast, our derivation, which leverages a general version of Tweedie's formula on high order moments of the posterior, can be extended to directly learning high order scores.

# B Proof

In the following, we assume that $q_\sigma(\tilde{\mathbf{x}}|\mathbf{x}) = \mathcal{N}(\tilde{\mathbf{x}}|\mathbf{x}, \sigma^2 I)$. Tweedie's formula can also be derived using the proof for Theorem 1.

**Theorem 1.** *Given D-dimensional densities $p(\mathbf{x})$ and $q_\sigma(\tilde{\mathbf{x}}) \triangleq \int p(\mathbf{x})q_\sigma(\tilde{\mathbf{x}}|\mathbf{x})d\mathbf{x}$, we have*

$$\mathbb{E}[\mathbf{x}\mathbf{x}^\mathsf{T} \mid \tilde{\mathbf{x}}] = \mathbf{f}(\tilde{\mathbf{x}}, \tilde{\mathbf{s}}_1, \tilde{\mathbf{s}}_2) \tag{21}$$

$$\mathbb{E}[\mathbf{x}\mathbf{x}^\mathsf{T} - \mathbf{x}\tilde{\mathbf{x}}^\mathsf{T} - \tilde{\mathbf{x}}\mathbf{x}^\mathsf{T} \mid \tilde{\mathbf{x}}] = \mathbf{h}(\tilde{\mathbf{x}}, \tilde{\mathbf{s}}_1, \tilde{\mathbf{s}}_2), \tag{22}$$

*where $\mathbf{f}(\tilde{\mathbf{x}}, \tilde{\mathbf{s}}_1, \tilde{\mathbf{s}}_2)$ and $\mathbf{h}(\tilde{\mathbf{x}}, \tilde{\mathbf{s}}_1, \tilde{\mathbf{s}}_2)$ are polynomials of $\tilde{\mathbf{x}}, \tilde{\mathbf{s}}_1(\tilde{\mathbf{x}}), \tilde{\mathbf{s}}_2(\tilde{\mathbf{x}})$ defined as*

$$\mathbf{f}(\tilde{\mathbf{x}}, \tilde{\mathbf{s}}_1, \tilde{\mathbf{s}}_2) = \tilde{\mathbf{x}}\tilde{\mathbf{x}}^\mathsf{T} + \sigma^2 \tilde{\mathbf{x}}\tilde{\mathbf{s}}_1(\tilde{\mathbf{x}})^\mathsf{T} + \sigma^2 \tilde{\mathbf{s}}_1(\tilde{\mathbf{x}})\tilde{\mathbf{x}}^\mathsf{T} + \sigma^4 \tilde{\mathbf{s}}_2(\tilde{\mathbf{x}}) + \sigma^4 \tilde{\mathbf{s}}_1(\tilde{\mathbf{x}})\tilde{\mathbf{s}}_1(\tilde{\mathbf{x}})^\mathsf{T} + \sigma^2 I, \tag{23}$$

$$\mathbf{h}(\tilde{\mathbf{x}}, \tilde{\mathbf{s}}_1, \tilde{\mathbf{s}}_2) = -\tilde{\mathbf{x}}\tilde{\mathbf{x}}^\mathsf{T} + \sigma^4 \tilde{\mathbf{s}}_2(\tilde{\mathbf{x}}) + \sigma^4 \tilde{\mathbf{s}}_1(\tilde{\mathbf{x}})\tilde{\mathbf{s}}_1(\tilde{\mathbf{x}})^\mathsf{T} + \sigma^2 I. \tag{24}$$

*Here $\tilde{\mathbf{s}}_1(\tilde{\mathbf{x}})$ and $\tilde{\mathbf{s}}_2(\tilde{\mathbf{x}})$ denote the first and second order scores of $q_\sigma(\tilde{\mathbf{x}})$.*

*Proof.* We can rewrite $q_\sigma(\tilde{\mathbf{x}}|\mathbf{x})$ in the form of exponential family

$$q_\sigma(\tilde{\mathbf{x}}|\boldsymbol{\eta}) = e^{\boldsymbol{\eta}^\mathsf{T}\tilde{\mathbf{x}} - \psi(\boldsymbol{\eta})} q_0(\tilde{\mathbf{x}}),$$

where $\boldsymbol{\eta} = \frac{\mathbf{x}}{\sigma^2}$ is the natural or canonical parameter of the family, $\psi(\boldsymbol{\eta})$ is the cumulant generating function which makes $q_\sigma(\tilde{\mathbf{x}}|\boldsymbol{\eta})$ normalized and $q_0(\tilde{\mathbf{x}}) = ((2\pi)^d \sigma^{2d})^{-\frac{1}{2}} e^{-\frac{\tilde{\mathbf{x}}^\mathsf{T}\tilde{\mathbf{x}}}{2\sigma^2}}$.

Bayes rule provides the corresponding posterior

$$q(\boldsymbol{\eta}|\tilde{\mathbf{x}}) = \frac{q_\sigma(\tilde{\mathbf{x}}|\boldsymbol{\eta})p(\boldsymbol{\eta})}{q_\sigma(\tilde{\mathbf{x}})}.$$

Let $\lambda(\tilde{\mathbf{x}}) = \log \frac{q_\sigma(\tilde{\mathbf{x}})}{q_0(\tilde{\mathbf{x}})}$, then we can write posterior as

$$q(\boldsymbol{\eta}|\tilde{\mathbf{x}}) = e^{\boldsymbol{\eta}^\mathsf{T}\tilde{\mathbf{x}} - \psi(\boldsymbol{\eta}) - \lambda(\tilde{\mathbf{x}})} p(\boldsymbol{\eta}).$$

Since the posterior is normalized, we have

$$\int e^{\boldsymbol{\eta}^\mathsf{T}\tilde{\mathbf{x}} - \psi(\boldsymbol{\eta}) - \lambda(\tilde{\mathbf{x}})} p(\boldsymbol{\eta})d\boldsymbol{\eta} = 1.$$

As a widely used technique in exponential families, we differentiate both sides w.r.t. $\tilde{\mathbf{x}}$

$$\int (\boldsymbol{\eta}^\mathsf{T} - \boldsymbol{J}_\lambda(\tilde{\mathbf{x}})^\mathsf{T}) q(\boldsymbol{\eta}|\tilde{\mathbf{x}})d\boldsymbol{\eta} = 0,$$

and the first order posterior moment can be written as

$$\mathbb{E}[\boldsymbol{\eta} \mid \tilde{\mathbf{x}}] = \boldsymbol{J}_\lambda(\tilde{\mathbf{x}}) \tag{25}$$

$$\mathbb{E}[\boldsymbol{\eta}^\mathsf{T} \mid \tilde{\mathbf{x}}] = \boldsymbol{J}_\lambda(\tilde{\mathbf{x}})^\mathsf{T}, \tag{26}$$

where $\boldsymbol{J}_\lambda(\tilde{\mathbf{x}})$ is the Jacobian of $\lambda(\tilde{\mathbf{x}})$ w.r.t. $\tilde{\mathbf{x}}$.

Differentiating both sides w.r.t. $\tilde{\mathbf{x}}$ again

$$\int \boldsymbol{\eta}(\boldsymbol{\eta}^\mathsf{T} - \boldsymbol{J}_\lambda(\tilde{\mathbf{x}})^\mathsf{T}) q(\boldsymbol{\eta} | \tilde{\mathbf{x}}) d\boldsymbol{\eta} = \boldsymbol{H}_\lambda(\tilde{\mathbf{x}}),$$

and the second order posterior moment can be written as

$$\mathbb{E}[\boldsymbol{\eta}\boldsymbol{\eta}^\mathsf{T} \mid \tilde{\mathbf{x}}] = \boldsymbol{H}_\lambda(\tilde{\mathbf{x}}) + \boldsymbol{J}_\lambda(\tilde{\mathbf{x}})\boldsymbol{J}_\lambda(\tilde{\mathbf{x}})^\mathsf{T}, \tag{27}$$

where $\boldsymbol{H}_\lambda(\tilde{\mathbf{x}})$ is the Hessian of $\lambda(\tilde{\mathbf{x}})$ w.r.t. $\tilde{\mathbf{x}}$.

Specifically, for $q_\sigma(\tilde{\mathbf{x}}|\mathbf{x}) = \mathcal{N}(\tilde{\mathbf{x}}|\mathbf{x}, \sigma^2 I)$, we have $\boldsymbol{\eta} = \frac{\mathbf{x}}{\sigma^2}$ and $q_0(\tilde{\mathbf{x}}) = ((2\pi)^d \sigma^{2d})^{-\frac{1}{2}} e^{-\frac{\tilde{\mathbf{x}}^\mathsf{T}\tilde{\mathbf{x}}}{2\sigma^2}}$.
Hence we have

$$\lambda(\tilde{\mathbf{x}}) = \log q_\sigma(\tilde{\mathbf{x}}) + \frac{\tilde{\mathbf{x}}^\mathsf{T}\tilde{\mathbf{x}}}{2\sigma^2} + \text{constant}$$

$$\boldsymbol{J}_\lambda(\tilde{\mathbf{x}}) = \tilde{\mathbf{s}}_1(\tilde{\mathbf{x}}) + \frac{\tilde{\mathbf{x}}}{\sigma^2}$$

$$\boldsymbol{H}_\lambda(\tilde{\mathbf{x}}) = \tilde{\mathbf{s}}_2(\tilde{\mathbf{x}}) + \frac{1}{\sigma^2} I.$$

From Eq. (27), we have

$$\mathbb{E}[\mathbf{x}\mathbf{x}^\mathsf{T} \mid \tilde{\mathbf{x}}] = \tilde{\mathbf{x}}\tilde{\mathbf{x}}^\mathsf{T} + \sigma^2 \tilde{\mathbf{x}}\tilde{\mathbf{s}}_1(\tilde{\mathbf{x}})^\mathsf{T} + \sigma^2 \tilde{\mathbf{s}}_1(\tilde{\mathbf{x}})\tilde{\mathbf{x}}^\mathsf{T} + \sigma^4 \tilde{\mathbf{s}}_2(\tilde{\mathbf{x}}) + \sigma^4 \tilde{\mathbf{s}}_1(\tilde{\mathbf{x}})\tilde{\mathbf{s}}_1(\tilde{\mathbf{x}})^\mathsf{T} + \sigma^2 I = \mathbf{f}(\tilde{\mathbf{x}}, \tilde{\mathbf{s}}_1, \tilde{\mathbf{s}}_2).$$

Combined with Eq. (25), Eq. (26), and Eq. (27), we have

$$\mathbb{E}[\mathbf{x}\mathbf{x}^\mathsf{T} - \mathbf{x}\tilde{\mathbf{x}}^\mathsf{T} - \tilde{\mathbf{x}}\mathbf{x}^\mathsf{T} \mid \tilde{\mathbf{x}}] = -\tilde{\mathbf{x}}\tilde{\mathbf{x}}^\mathsf{T} + \sigma^4 \tilde{\mathbf{s}}_2(\tilde{\mathbf{x}}) + \sigma^4 \tilde{\mathbf{s}}_1(\tilde{\mathbf{x}})\tilde{\mathbf{s}}_1(\tilde{\mathbf{x}})^\mathsf{T} + \sigma^2 I = \mathbf{h}(\tilde{\mathbf{x}}, \tilde{\mathbf{s}}_1, \tilde{\mathbf{s}}_2).$$

$\square$

**Tweedie's formula.** *Given D-dimensional densities $p(\mathbf{x})$ and $q_\sigma(\tilde{\mathbf{x}}) \triangleq \int p(\mathbf{x}) q_\sigma(\tilde{\mathbf{x}}|\mathbf{x}) d\mathbf{x}$, we have*

$$\mathbb{E}[\mathbf{x} \mid \tilde{\mathbf{x}}] = \tilde{\mathbf{x}} + \sigma^2 \tilde{\mathbf{s}}_1(\tilde{\mathbf{x}}), \tag{28}$$

*where $\tilde{\mathbf{s}}_1(\tilde{\mathbf{x}}) \triangleq \nabla_{\tilde{\mathbf{x}}} \log q_\sigma(\tilde{\mathbf{x}})$.*

*Proof.* Plug in $\boldsymbol{\eta} = \frac{\mathbf{x}}{\sigma^2}$ and $\boldsymbol{J}_\lambda(\tilde{\mathbf{x}}) = \tilde{\mathbf{s}}_1(\tilde{\mathbf{x}}) + \frac{\tilde{\mathbf{x}}}{\sigma^2}$ in Eq. (25), we have

$$\mathbb{E}[\mathbf{x} \mid \tilde{\mathbf{x}}] = \tilde{\mathbf{x}} + \sigma^2 \tilde{\mathbf{s}}_1(\tilde{\mathbf{x}}), \tag{29}$$

which proves Tweedie's formula. $\square$

**Theorem 2.** *Suppose the first order score $\tilde{\mathbf{s}}_1(\tilde{\mathbf{x}})$ is given, we can learn a second order score model $\tilde{\mathbf{s}}_2(\tilde{\mathbf{x}}; \boldsymbol{\theta})$ by optimizing the following objectives*

$$\boldsymbol{\theta}^* = \arg\min_{\boldsymbol{\theta}} \mathbb{E}_{p_{data}(\mathbf{x})} \mathbb{E}_{q_\sigma(\tilde{\mathbf{x}}|\mathbf{x})} \left[ \left\| \mathbf{x}\mathbf{x}^\mathsf{T} - \mathbf{f}(\tilde{\mathbf{x}}, \tilde{\mathbf{s}}_1(\tilde{\mathbf{x}}), \tilde{\mathbf{s}}_2(\tilde{\mathbf{x}}; \boldsymbol{\theta})) \right\|_2^2 \right],$$

$$\boldsymbol{\theta}^* = \arg\min_{\boldsymbol{\theta}} \mathbb{E}_{p_{data}(\mathbf{x})} \mathbb{E}_{q_\sigma(\tilde{\mathbf{x}}|\mathbf{x})} \left[ \left\| \mathbf{x}\mathbf{x}^\mathsf{T} - \mathbf{x}\tilde{\mathbf{x}}^\mathsf{T} - \tilde{\mathbf{x}}\mathbf{x}^\mathsf{T} - \mathbf{h}(\tilde{\mathbf{x}}, \tilde{\mathbf{s}}_1(\tilde{\mathbf{x}}), \tilde{\mathbf{s}}_2(\tilde{\mathbf{x}}; \boldsymbol{\theta})) \right\|_2^2 \right]$$

*where $\mathbf{f}(\cdot)$ and $\mathbf{h}(\cdot)$ are polynomials defined in Eq. (9) and Eq. (10). Assuming the model has an infinite capacity, then the optimal parameter $\boldsymbol{\theta}^*$ satisfies $\tilde{\mathbf{s}}_2(\tilde{\mathbf{x}}; \boldsymbol{\theta}^*) = \tilde{\mathbf{s}}_2(\tilde{\mathbf{x}})$ for almost any $\tilde{\mathbf{x}}$.*

*Proof.* It is well-known that the optimal solution to the least squares regression problems of Eq. (11) and Eq. (12) are the conditional expectations $\mathbf{f}(\tilde{\mathbf{x}}, \tilde{\mathbf{s}}_1(\tilde{\mathbf{x}}), \tilde{\mathbf{s}}_2(\tilde{\mathbf{x}}; \boldsymbol{\theta}^*)) = \mathbb{E}[\mathbf{x}\mathbf{x}^\mathsf{T} \mid \tilde{\mathbf{x}}]$ and $\mathbf{h}(\tilde{\mathbf{x}}, \tilde{\mathbf{s}}_1(\tilde{\mathbf{x}}), \tilde{\mathbf{s}}_2(\tilde{\mathbf{x}}; \boldsymbol{\theta}^*)) = \mathbb{E}[\mathbf{x}\mathbf{x}^\mathsf{T} - \mathbf{x}\tilde{\mathbf{x}}^\mathsf{T} - \tilde{\mathbf{x}}\mathbf{x}^\mathsf{T} \mid \tilde{\mathbf{x}}]$ respectively. According to Theorem 1, this implies that the optimal solution satisfies $\tilde{\mathbf{s}}_2(\tilde{\mathbf{x}}; \boldsymbol{\theta}^*) = \tilde{\mathbf{s}}_2(\tilde{\mathbf{x}})$ for almost any $\tilde{\mathbf{x}}$ given the first order score $\tilde{\mathbf{s}}_1(\tilde{\mathbf{x}})$.

**Note:** Eq. (11) and Eq. (12) have the same set of solutions assuming sufficient model capacity. However, Eq. (12) has a simpler form (e.g., involving fewer terms) than Eq. (11) since multiple terms in Eq. (12) can be cancelled after expanding the equation by using Eq. (4) (Tweedie's formula), resulting in the simplified objective Eq. (13). Compared to the expansion of Eq. (11), the expansion of Eq. (12) (i.e., Eq. (13)) is much simpler (i.e., involving fewer terms), which is why we use Eq. (12) other than Eq. (11) in our experiments. □

Before proving Theorem 3, we first prove the following lemma.

**Lemma 1.** *Given a $D$ dimensional distribution $p_{data}(\mathbf{x})$, and $q_\sigma(\tilde{\mathbf{x}}|\mathbf{x}) \triangleq \mathcal{N}(\tilde{\mathbf{x}}|\mathbf{x}, \sigma^2 I)$, we have the following for any integer $n \geqslant 1$:*

$$\mathbb{E}[\otimes^{n+1}\mathbf{x}|\tilde{\mathbf{x}}] = \sigma^2 \frac{\partial}{\partial \tilde{\mathbf{x}}}\mathbb{E}[\otimes^n\mathbf{x}|\tilde{\mathbf{x}}] + \sigma^2 \mathbb{E}[\otimes^n\mathbf{x}|\tilde{\mathbf{x}}] \otimes \left(\tilde{\mathbf{s}}_1(\tilde{\mathbf{x}}) + \frac{\tilde{\mathbf{x}}}{\sigma^2}\right),$$

*where $\otimes^n\mathbf{x} \in \mathbb{R}^{D^n}$ denotes $n$-fold tensor multiplications.*

*Proof.* We follow the notation used in the previous proof. Since

$$\mathbb{E}[\otimes^n\boldsymbol{\eta}|\tilde{\mathbf{x}}] = \int e^{\boldsymbol{\eta}^\mathsf{T}\tilde{\mathbf{x}} - \psi(\boldsymbol{\eta}) - \lambda(\tilde{\mathbf{x}})} p(\boldsymbol{\eta}) \otimes^n \boldsymbol{\eta} d\boldsymbol{\eta},$$

differentiating both sides w.r.t. $\tilde{\mathbf{x}}$

$$\frac{\partial}{\partial \tilde{\mathbf{x}}}\mathbb{E}[\otimes^n\boldsymbol{\eta}|\tilde{\mathbf{x}}] = \int e^{\boldsymbol{\eta}^\mathsf{T}\tilde{\mathbf{x}} - \psi(\boldsymbol{\eta}) - \lambda(\tilde{\mathbf{x}})} p(\boldsymbol{\eta}) \otimes^{n+1} \boldsymbol{\eta} d\boldsymbol{\eta} - \int e^{\boldsymbol{\eta}^\mathsf{T}\tilde{\mathbf{x}} - \psi(\boldsymbol{\eta}) - \lambda(\tilde{\mathbf{x}})} p(\boldsymbol{\eta}) \otimes^n \boldsymbol{\eta} d\boldsymbol{\eta} \otimes \frac{\partial}{\partial \tilde{\mathbf{x}}}\lambda(\tilde{\mathbf{x}})$$

$$\frac{\partial}{\partial \tilde{\mathbf{x}}}\mathbb{E}[\otimes^n\boldsymbol{\eta}|\tilde{\mathbf{x}}] = \mathbb{E}[\otimes^{n+1}\boldsymbol{\eta}|\tilde{\mathbf{x}}] - \mathbb{E}[\otimes^n\boldsymbol{\eta}|\tilde{\mathbf{x}}] \otimes \left(\tilde{\mathbf{s}}_1(\tilde{\mathbf{x}}) + \frac{\tilde{\mathbf{x}}}{\sigma^2}\right).$$

Thus

$$\mathbb{E}[\otimes^{n+1}\mathbf{x}|\tilde{\mathbf{x}}] = \sigma^2 \frac{\partial}{\partial \tilde{\mathbf{x}}}\mathbb{E}[\otimes^n\mathbf{x}|\tilde{\mathbf{x}}] + \sigma^2 \mathbb{E}[\otimes^n\mathbf{x}|\tilde{\mathbf{x}}] \otimes \left(\tilde{\mathbf{s}}_1(\tilde{\mathbf{x}}) + \frac{\tilde{\mathbf{x}}}{\sigma^2}\right).$$

□

**Example** When $n = 2$, plug in Eq. (4), we have

$$\mathbb{E}[\mathbf{x}\mathbf{x}^\mathsf{T}|\tilde{\mathbf{x}}] = \sigma^2(I + \sigma^2\tilde{\mathbf{s}}_2(\tilde{\mathbf{x}})) + \sigma^2(\tilde{\mathbf{x}} + \sigma^2\tilde{\mathbf{s}}_1(\tilde{\mathbf{x}}))(\tilde{\mathbf{s}}_1(\tilde{\mathbf{x}}) + \frac{\tilde{\mathbf{x}}}{\sigma^2})^\mathsf{T},$$

which can be simplified as Eq. (9).

Lemma 1 provides a recurrence for obtaining $\mathbf{f}_n$ in closed form. It is further used and discussed in Theorem 3.

**Theorem 3.** $\mathbb{E}[\otimes^n\mathbf{x}|\tilde{\mathbf{x}}] = \mathbf{f}_n(\tilde{\mathbf{x}}, \tilde{\mathbf{s}}_1, ..., \tilde{\mathbf{s}}_n)$, *where $\otimes^n\mathbf{x} \in \mathbb{R}^{D^n}$ denotes $n$-fold tensor multiplications, $\mathbf{f}_n(\tilde{\mathbf{x}}, \tilde{\mathbf{s}}_1, ..., \tilde{\mathbf{s}}_n)$ is a polynomial of $\{\tilde{\mathbf{x}}, \tilde{\mathbf{s}}_1(\tilde{\mathbf{x}}), ..., \tilde{\mathbf{s}}_n(\tilde{\mathbf{x}})\}$ and $\tilde{\mathbf{s}}_k(\mathbf{x})$ represents the $k$-th order score of $q_\sigma(\tilde{\mathbf{x}}) = \int p_{data}(\mathbf{x})q_\sigma(\tilde{\mathbf{x}}|\mathbf{x})d\mathbf{x}$.*

*Proof.* We prove this using induction. When $n = 1$, we have

$$\mathbb{E}[\mathbf{x}|\tilde{\mathbf{x}}] = \sigma^2\tilde{\mathbf{s}}_1(\tilde{\mathbf{x}}) + \tilde{\mathbf{x}}.$$

Thus, $\mathbb{E}[\mathbf{x}|\tilde{\mathbf{x}}]$ can be written as a polynomial of $\{\tilde{\mathbf{x}}, \tilde{\mathbf{s}}_1(\tilde{\mathbf{x}})\}$. The hypothesis holds.

Assume the hypothesis holds when $n = t$, then

$$\mathbb{E}[\otimes^t\mathbf{x}|\tilde{\mathbf{x}}] = \mathbf{f}_t(\tilde{\mathbf{x}}, \tilde{\mathbf{s}}_1, ..., \tilde{\mathbf{s}}_t).$$

When $n = t + 1$,

$$\mathbb{E}[\otimes^{t+1}\mathbf{x}|\tilde{\mathbf{x}}] = \sigma^2 \frac{\partial}{\partial \tilde{\mathbf{x}}} \mathbb{E}[\otimes^t \mathbf{x}|\tilde{\mathbf{x}}] + \sigma^2 \mathbb{E}[\otimes^t \mathbf{x}|\tilde{\mathbf{x}}] \otimes \left( \tilde{\mathbf{s}}_1(\tilde{\mathbf{x}}) + \frac{\tilde{\mathbf{x}}}{\sigma^2} \right)$$

$$= \sigma^2 \frac{\partial}{\partial \tilde{\mathbf{x}}} \mathbf{f}_t(\tilde{\mathbf{x}}, \tilde{\mathbf{s}}_1, ..., \tilde{\mathbf{s}}_t) + \sigma^2 \mathbf{f}_t(\tilde{\mathbf{x}}, \tilde{\mathbf{s}}_1, ..., \tilde{\mathbf{s}}_t) \otimes \left( \tilde{\mathbf{s}}_1(\tilde{\mathbf{x}}) + \frac{\tilde{\mathbf{x}}}{\sigma^2} \right).$$

Clearly, $\sigma^2 \mathbf{f}_t(\tilde{\mathbf{x}}, \tilde{\mathbf{s}}_1, ..., \tilde{\mathbf{s}}_t) \otimes \left( \tilde{\mathbf{s}}_1(\tilde{\mathbf{x}}) + \frac{\tilde{\mathbf{x}}}{\sigma^2} \right)$ is a polynomial of $\{\tilde{\mathbf{x}}, \tilde{\mathbf{s}}_1(\tilde{\mathbf{x}}), ..., \tilde{\mathbf{s}}_t(\tilde{\mathbf{x}})\}$, and $\sigma^2 \frac{\partial}{\partial \tilde{\mathbf{x}}} \mathbf{f}_t(\tilde{\mathbf{x}}, \tilde{\mathbf{s}}_1, ..., \tilde{\mathbf{s}}_t)$ is a polynomial of $\{\tilde{\mathbf{x}}, \tilde{\mathbf{s}}_1(\tilde{\mathbf{x}}), ..., \tilde{\mathbf{s}}_{t+1}(\tilde{\mathbf{x}})\}$. This implies $\mathbb{E}[\otimes^{t+1}\mathbf{x}|\tilde{\mathbf{x}}]$ can be written as $\mathbf{f}_{t+1}(\tilde{\mathbf{x}}, \tilde{\mathbf{s}}_1, ..., \tilde{\mathbf{s}}_{t+1})$, which is a polynomial of $\{\tilde{\mathbf{x}}, \tilde{\mathbf{s}}_1(\tilde{\mathbf{x}}), ..., \tilde{\mathbf{s}}_{t+1}(\tilde{\mathbf{x}})\}$. Thus, the hypothesis holds when $k = t + 1$, which implies that the hypothesis holds for all integer $n \geqslant 1$. $\square$

**Theorem 4.** *Given the true score functions $\tilde{\mathbf{s}}_1(\tilde{\mathbf{x}}), ..., \tilde{\mathbf{s}}_{k-1}(\tilde{\mathbf{x}})$, a k-th order score model $\tilde{\mathbf{s}}_k(\tilde{\mathbf{x}}; \boldsymbol{\theta})$, and*

$$\boldsymbol{\theta}^* = \arg\min_{\boldsymbol{\theta}} \mathbb{E}_{p_{data}(\mathbf{x})} \mathbb{E}_{q_\sigma(\tilde{\mathbf{x}}|\mathbf{x})} [\| \otimes^k \mathbf{x} - \mathbf{f}_k(\tilde{\mathbf{x}}, \tilde{\mathbf{s}}_1(\tilde{\mathbf{x}}), ..., \tilde{\mathbf{s}}_{k-1}(\tilde{\mathbf{x}}), \tilde{\mathbf{s}}_k(\tilde{\mathbf{x}}; \boldsymbol{\theta}))\|^2]. \tag{30}$$

*Assuming the model has an infinite capacity, we have $\tilde{\mathbf{s}}_k(\tilde{\mathbf{x}}; \boldsymbol{\theta}^*) = \tilde{\mathbf{s}}_k(\tilde{\mathbf{x}})$ for almost all $\tilde{\mathbf{x}}$.*

*Proof.* Similar to the previous case, we can show that the solution to the least squares regression problems of Eq. (30) is $\mathbf{f}_k(\tilde{\mathbf{x}}, \tilde{\mathbf{s}}_1(\tilde{\mathbf{x}}), ..., \tilde{\mathbf{s}}_{k-1}(\tilde{\mathbf{x}}), \tilde{\mathbf{s}}_k(\tilde{\mathbf{x}}; \boldsymbol{\theta})^*) = \mathbb{E}[\otimes^t \mathbf{x}|\tilde{\mathbf{x}}]$. According to Theorem 3, this implies $\tilde{\mathbf{s}}_k(\tilde{\mathbf{x}}; \boldsymbol{\theta}^*) = \tilde{\mathbf{s}}_k(\tilde{\mathbf{x}})$ given the score functions $\tilde{\mathbf{s}}_1(\tilde{\mathbf{x}}), ..., \tilde{\mathbf{s}}_{k-1}(\tilde{\mathbf{x}})$. $\square$

## C Analysis on Second Order Score Models

### C.1 Variance reduction

If we want to match the score of true distribution $p_{\text{data}}(\mathbf{x})$, $\sigma$ should be approximately zero for both DSM and $D_2$SM so that $q_\sigma(\tilde{\mathbf{x}})$ is close to $p_{\text{data}}(\mathbf{x})$. However, when $\sigma \to 0$, both DSM and $D_2$SM can be unstable to train and might not converge, which calls for variance reduction techniques. In this section, we show that we can introduce a control variate to improve the empirical performance of DSM and $D_2$SM when $\sigma$ tends to zero. Our variance control method can be derived from expanding the original training objective function using Taylor expansion.

**DSM with varaince reduction** Expand the objective using Taylor expansion

$$\mathcal{L}_{DSM}(\boldsymbol{\theta}) = \frac{1}{2} \mathbb{E}_{p_{\text{data}}(\mathbf{x})} \mathbb{E}_{q_\sigma(\tilde{\mathbf{x}}|\mathbf{x})} \left[ \left\| \tilde{\mathbf{s}}_1(\tilde{\mathbf{x}}; \boldsymbol{\theta}) + \frac{1}{\sigma^2}(\tilde{\mathbf{x}} - \mathbf{x}) \right\|_2^2 \right]$$

$$= \frac{1}{2} \mathbb{E}_{p_{\text{data}}(\mathbf{x})} \mathbb{E}_{\mathbf{z} \sim \mathcal{N}(0, I)} \left[ \left\| \tilde{\mathbf{s}}_1(\mathbf{x} + \sigma \mathbf{z}; \boldsymbol{\theta}) + \frac{\mathbf{z}}{\sigma} \right\|_2^2 \right]$$

$$= \frac{1}{2} \mathbb{E}_{p_{\text{data}}(\mathbf{x})} \mathbb{E}_{\mathbf{z} \sim \mathcal{N}(0, I)} \left[ \| \tilde{\mathbf{s}}_1(\mathbf{x} + \sigma \mathbf{z}; \boldsymbol{\theta}) \|_2^2 + \frac{2}{\sigma} \tilde{\mathbf{s}}_1(\mathbf{x} + \sigma \mathbf{z}; \boldsymbol{\theta})^T \mathbf{z} + \frac{\|\mathbf{z}\|_2^2}{\sigma^2} \right]$$

$$= \frac{1}{2} \mathbb{E}_{p_{\text{data}}(\mathbf{x})} \mathbb{E}_{\mathbf{z} \sim \mathcal{N}(0, I)} \left[ \| \tilde{\mathbf{s}}_1(\mathbf{x}; \boldsymbol{\theta}) \|_2^2 + \frac{2}{\sigma} \tilde{\mathbf{s}}_1(\mathbf{x}; \boldsymbol{\theta})^T \mathbf{z} + \frac{\|\mathbf{z}\|_2^2}{\sigma^2} \right] + \mathcal{O}(1),$$

where $\mathcal{O}(1)$ is bounded when $\sigma \to 0$. Since

$$\mathbb{E}_{\mathbf{z} \sim \mathcal{N}(0, I)} \left[ \frac{2}{\sigma} \tilde{\mathbf{s}}_1(\mathbf{x}; \boldsymbol{\theta})^T \mathbf{z} + \frac{\|\mathbf{z}\|_2^2}{\sigma^2} \right] = \frac{D}{\sigma^2}, \tag{31}$$

where $D$ is the dimension of $p_{\text{data}}(\mathbf{x})$, we can use Eq. (31) as a control variate and define DSM with variance reduction as

$$\mathcal{L}_{DSM-VR} = \mathcal{L}_{DSM} - \mathbb{E}_{p_{\text{data}}(\mathbf{x})} \mathbb{E}_{\mathbf{z} \sim \mathcal{N}(0, I)} \left[ \frac{2}{\sigma} \tilde{\mathbf{s}}_1(\mathbf{x}; \boldsymbol{\theta})^T \mathbf{z} + \frac{\|\mathbf{z}\|_2^2}{\sigma^2} \right] + \frac{D}{\sigma^2} \tag{32}$$

An equivalent version of Eq. (32) is first proposed in [30].

**$D_2$SM with variance reduction** We now derive the variance reduction objective for $D_2$SM. Let us first consider the $ij$-th term of $\mathcal{L}_{D_2\text{SM}}(\boldsymbol{\theta})$. We denote $\boldsymbol{\psi}(\tilde{\mathbf{x}};\boldsymbol{\theta}) = \tilde{\mathbf{s}}_2(\tilde{\mathbf{x}};\boldsymbol{\theta}) + \tilde{\mathbf{s}}_1(\tilde{\mathbf{x}};\boldsymbol{\theta})\tilde{\mathbf{s}}_1(\tilde{\mathbf{x}};\boldsymbol{\theta})^\mathsf{T}$ and $\boldsymbol{\psi}_{ij}(\tilde{\mathbf{x}};\boldsymbol{\theta})$ the $ij$-th term of $\boldsymbol{\psi}(\tilde{\mathbf{x}};\boldsymbol{\theta})$. Similar as the variance reduction method for DSM [30], we expand the objective of $D_2$SM (Eq. (13)) using Taylor expansion

$$\mathcal{L}_{D_2\text{SM}}(\boldsymbol{\theta})_{ij} = \frac{1}{2}\mathbb{E}_{p_{\text{data}}(\mathbf{x})}\mathbb{E}_{\mathbf{z}\sim\mathcal{N}(0,I)}[\boldsymbol{\psi}_{ij}(\mathbf{x}+\sigma\mathbf{z};\boldsymbol{\theta}) + \frac{\mathbf{I}_{ij}-z_iz_j}{\sigma^2}]^2$$

$$= \frac{1}{2}\mathbb{E}_{p_{\text{data}}(\mathbf{x})}\mathbb{E}_{\mathbf{z}\sim\mathcal{N}(0,I)}[\boldsymbol{\psi}_{ij}(\mathbf{x}+\sigma\mathbf{z};\boldsymbol{\theta})^2 + 2\frac{\mathbf{I}_{ij}-z_iz_j}{\sigma^2}\boldsymbol{\psi}_{ij}(\mathbf{x}+\sigma\mathbf{z};\boldsymbol{\theta}) + \frac{(\mathbf{I}_{ij}-z_iz_j)^2}{\sigma^4}]$$

$$= \frac{1}{2}\mathbb{E}_{p_{\text{data}}(\mathbf{x})}\mathbb{E}_{\mathbf{z}\sim\mathcal{N}(0,I)}[\boldsymbol{\psi}_{ij}(\mathbf{x};\boldsymbol{\theta})^2 + 2\frac{\mathbf{I}_{ij}-z_iz_j}{\sigma^2}\boldsymbol{\psi}_{ij}(\mathbf{x};\boldsymbol{\theta}) + 2\frac{\mathbf{I}_{ij}-z_iz_j}{\sigma}\mathbf{J}_{\boldsymbol{\psi}_{ij}}\mathbf{z} + \frac{(\mathbf{I}_{ij}-z_iz_j)^2}{\sigma^4}] + \mathcal{O}(1),$$

where $\mathcal{O}(1)$ is bounded when $\sigma \to 0$. It is clear to see that the term $\frac{(\mathbf{I}_{ij}-z_iz_j)^2}{\sigma^4}$ is a constant w.r.t. optimization. When $\sigma$ approximates zero, both $\frac{\mathbf{I}_{ij}-z_iz_j}{\sigma^2}$ and $\frac{\mathbf{I}_{ij}-z_iz_j}{\sigma}$ would be very large, making the training process unstable and hard to converge. Thus we aim at designing a control variate to cancel out these two terms. To do this, we propose to use antithetic sampling. Instead of using independent noise samples, we use two correlated (opposite) noise vectors centered at $\mathbf{x}$ defined as $\tilde{\mathbf{x}}_+ = \tilde{\mathbf{x}} + \sigma\mathbf{z}$ and $\tilde{\mathbf{x}}_- = \tilde{\mathbf{x}} - \sigma\mathbf{z}$. We propose the following objective function to reduce variance

$$\mathcal{L}_{D_2\text{SM-VR}} = \mathbb{E}_{\mathbf{x}\sim p_{\text{data}}(\mathbf{x})}\mathbb{E}_{\mathbf{z}\sim\mathcal{N}(0,I)}\left[\boldsymbol{\psi}(\tilde{\mathbf{x}}_+;\boldsymbol{\theta})^2 + \boldsymbol{\psi}(\tilde{\mathbf{x}}_-;\boldsymbol{\theta})^2 + 2\frac{\mathbf{I}-\mathbf{z}\mathbf{z}^\mathsf{T}}{\sigma}\odot(\boldsymbol{\psi}(\tilde{\mathbf{x}}_+;\boldsymbol{\theta}) + \boldsymbol{\psi}(\tilde{\mathbf{x}}_-;\boldsymbol{\theta}) - 2\boldsymbol{\psi}(\mathbf{x};\boldsymbol{\theta}))\right]. \quad (33)$$

Similarly, we can show easily by using Taylor expansion that optimizing Eq. (33) is equivalent to optimizing Eq. (13) up to a control variate. On the other hand, Eq. (33) is more stable to optimize than Eq. (13) when $\sigma$ approximates zero since the unstable terms $\frac{\mathbf{I}_{ij}-z_iz_j}{\sigma^2}$ and $\frac{\mathbf{I}_{ij}-z_iz_j}{\sigma}$ are both cancelled by the introduced control variate.

## C.2 Learning accuracy

This section provides more experimental details on Section 4.4. We use a 3-layer MLP model with latent size 128 and Tanh activation function for $\tilde{\mathbf{s}}_1(\tilde{\mathbf{x}};\boldsymbol{\theta})$. As discussed in Section 4.2, our $\tilde{\mathbf{s}}_2(\tilde{\mathbf{x}};\boldsymbol{\theta})$ model consists of two parts $\boldsymbol{\alpha}(\tilde{\mathbf{x}};\boldsymbol{\theta})$ and $\boldsymbol{\beta}(\tilde{\mathbf{x}};\boldsymbol{\theta})$. We also use a 3-layer MLP model with latent size 32 and Tanh activation function to parameterize $\boldsymbol{\alpha}(\tilde{\mathbf{x}};\boldsymbol{\theta})$ and $\boldsymbol{\beta}(\tilde{\mathbf{x}};\boldsymbol{\theta})$. For the mean squared error diagonal comparison experiments, we only parameterize the diagonal component $\boldsymbol{\alpha}(\tilde{\mathbf{x}};\boldsymbol{\theta})$. We use a 3-layer MLP model with latent size 32, and Tanh activation function to parameterize $\boldsymbol{\alpha}(\tilde{\mathbf{x}};\boldsymbol{\theta})$. We use learning rate 0.001, and train the models using Adam optimizer until convergence. We use noise scale $\sigma = 0.01, 0.05, 0.1$ during training so that the noise perturbed distribution $q_\sigma(\tilde{\mathbf{x}})$ is close to $p_{\text{data}}(\mathbf{x})$. All the mean squared error results in Table 1 are computed w.r.t. to the ground truth second order score of the clean data $p_{\text{data}}(\mathbf{x})$. The experiments are performed on 1 GPU.

## C.3 Computational efficiency

This section provides more experimental details on the computational efficiency experiments in Section 4.4. In the experiment, we consider two types of models.

**MLP model** We use a 3-layer MLP model to parameterize $\tilde{\mathbf{s}}_1(\tilde{\mathbf{x}};\boldsymbol{\theta})$ for a 100 dimensional data distribution. As discussed in Section 4.2, our $\tilde{\mathbf{s}}_2(\tilde{\mathbf{x}};\boldsymbol{\theta})$ model consists of two parts $\boldsymbol{\alpha}(\tilde{\mathbf{x}};\boldsymbol{\theta})$ and $\boldsymbol{\beta}(\tilde{\mathbf{x}};\boldsymbol{\theta})$. We use a 3-layer MLP model with comparable amount of parameters as $\tilde{\mathbf{s}}_1(\tilde{\mathbf{x}};\boldsymbol{\theta})$ to parameterize $\boldsymbol{\alpha}(\tilde{\mathbf{x}};\boldsymbol{\theta})$ and $\boldsymbol{\beta}(\tilde{\mathbf{x}};\boldsymbol{\theta})$. We consider rank $r = 20, 50, 200$ and $1000$ for $\boldsymbol{\beta}(\tilde{\mathbf{x}};\boldsymbol{\theta})$ in the experiment as reported in Table 2.

**U-Net model** We use a U-Net model to parameterize $\tilde{\mathbf{s}}_1(\tilde{\mathbf{x}};\boldsymbol{\theta})$ for the 784 dimensional data distribution. We use a similar U-Net architecture as $\tilde{\mathbf{s}}_1(\tilde{\mathbf{x}};\boldsymbol{\theta})$ for parameterizing $\boldsymbol{\alpha}(\tilde{\mathbf{x}};\boldsymbol{\theta})$ and $\boldsymbol{\beta}(\tilde{\mathbf{x}};\boldsymbol{\theta})$, except that we modify the output channel size to match the rank $r$ of $\boldsymbol{\beta}(\tilde{\mathbf{x}};\boldsymbol{\theta})$. We consider rank $r = 20, 50, 200$ and $1000$ for $\boldsymbol{\beta}(\tilde{\mathbf{x}};\boldsymbol{\theta})$ in the experiment as reported in Table 2. All the experiments are performed on the same TITAN Xp GPU using exactly the same computational setting. We use the implementation of U-Net from this repository `https://github.com/ermongroup/ncsn`.

# D Uncertainty Quantification

This section provides more experimental details on Section 5.

### D.1 Synthetic experiments

This section provides more details on the synthetic data experiments. We use a 3-layer MLP model for both $\tilde{\mathbf{s}}_1(\tilde{\mathbf{x}}; \boldsymbol{\theta})$ and $\tilde{\mathbf{s}}_2(\tilde{\mathbf{x}}; \boldsymbol{\theta})$. We train $\tilde{\mathbf{s}}_1(\tilde{\mathbf{x}}; \boldsymbol{\theta})$ and $\tilde{\mathbf{s}}_2(\tilde{\mathbf{x}}; \boldsymbol{\theta})$ jointly with Eq. (15). We use $\sigma = 0.15$ for $q_\sigma(\tilde{\mathbf{x}}|\mathbf{x})$, and train the models using Adam optimizer until convergence. We observe that training $\tilde{\mathbf{s}}_1(\tilde{\mathbf{x}}; \boldsymbol{\theta})$ directly with DSM and training $\tilde{\mathbf{s}}_1(\tilde{\mathbf{x}}; \boldsymbol{\theta})$ jointly with Eq. (15) have the same empirical performance in terms of estimating $\tilde{\mathbf{s}}_1$. Thus, we train $\tilde{\mathbf{s}}_1(\tilde{\mathbf{x}}; \boldsymbol{\theta})$ jointly with $\tilde{\mathbf{s}}_2(\tilde{\mathbf{x}}; \boldsymbol{\theta})$ in our experiments.

### D.2 Convariance diagonal visualizations

For both the MNIST and CIFAR-10 models, we use U-Net architectures to parameterize $\tilde{\mathbf{s}}_1(\tilde{\mathbf{x}}; \boldsymbol{\theta})$. We also use a similar U-Net architecture to parameterize $\tilde{\mathbf{s}}_2(\tilde{\mathbf{x}}; \boldsymbol{\theta})$, except that we modify the output channel size to match the rank $r$ of $\boldsymbol{\beta}(\tilde{\mathbf{x}}; \boldsymbol{\theta})$. We use $r = 50$ for $\boldsymbol{\beta}(\tilde{\mathbf{x}}; \boldsymbol{\theta})$ for both MNIST and CIFAR-10 models. We use the U-Net model implementation from this repository `https://github.com/ermongroup/ncsn`. We consider noise scales $\sigma = 0.3, 0.5, 0.8, 1.0$ for MNIST and $\sigma = 0.3, 0.5, 0.8$ for CIFAR-10. We train the models jointly until convergence with Eq. (15), using learning rate 0.0002 with Adam optimizer. The models are trained on the corresponding training sets on 2 GPUs.

### D.3 Full convariance visualizations

We use U-Net architectures to parameterize $\tilde{\mathbf{s}}_1(\tilde{\mathbf{x}}; \boldsymbol{\theta})$. We also use a similar U-Net architecture to parameterize $\tilde{\mathbf{s}}_2(\tilde{\mathbf{x}}; \boldsymbol{\theta})$, except that we modify the output channel size to match the rank $r$ of $\boldsymbol{\beta}(\tilde{\mathbf{x}}; \boldsymbol{\theta})$. We use $r = 50$ for $\boldsymbol{\beta}(\tilde{\mathbf{x}}; \boldsymbol{\theta})$ for this experiment. We use the U-Net model implementation from this repository `https://github.com/ermongroup/ncsn`. We train the models until convergence, using learning rate 0.0002 with Adam optimizer. The models are trained on the corresponding training set on 2 GPUs. We provide extra eigenvector visualizations for Fig. 4 in Figs. 9 and 10.

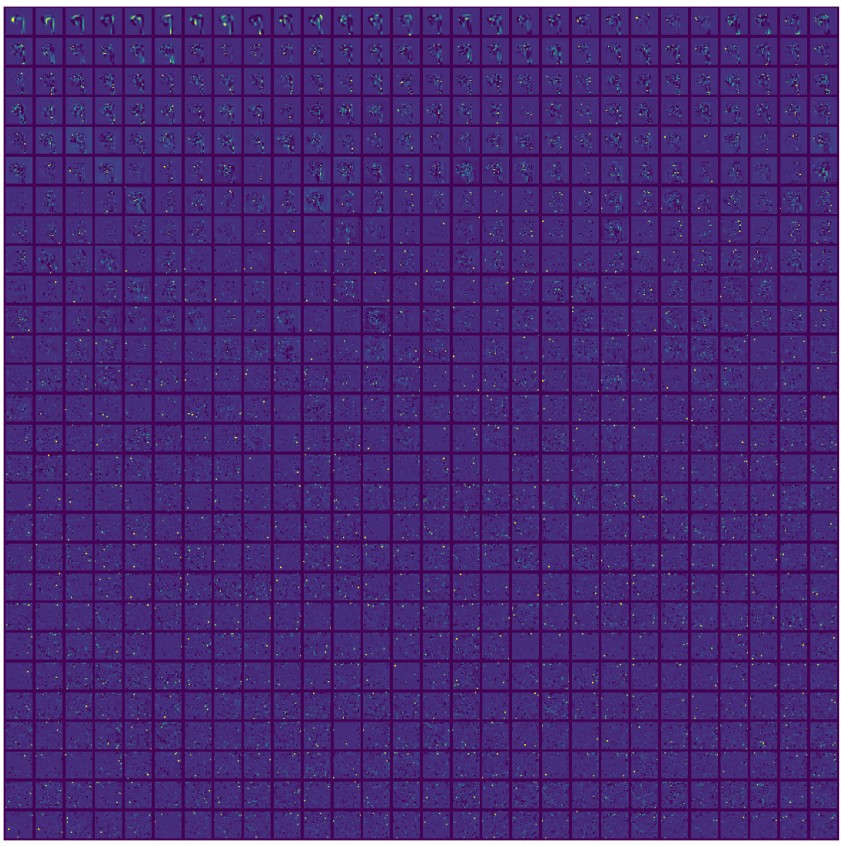

Figure 9: Eigenvectors (sorted by eigenvalues) of $\mathrm{Cov}[\mathbf{x} \mid \tilde{\mathbf{x}}]$ estimated by $\tilde{\mathbf{s}}_2(\tilde{\mathbf{x}}; \boldsymbol{\theta})$ on MNIST (more details in Section 5).

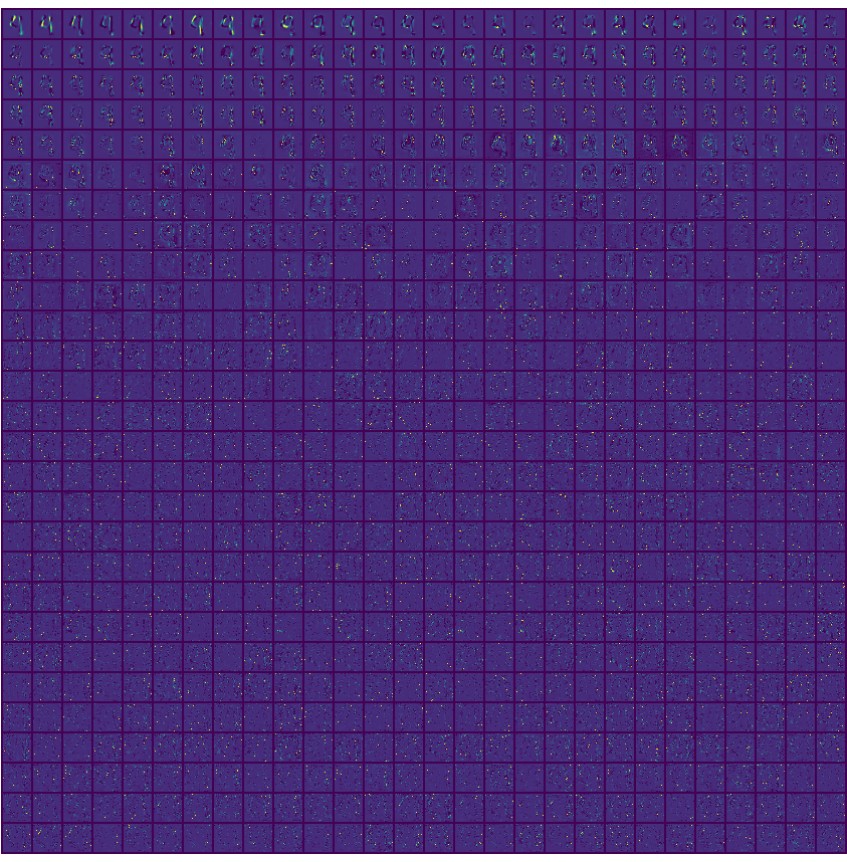

Figure 10: Eigenvectors (sorted by eigenvalues) of $\text{Cov}[\mathbf{x} \mid \tilde{\mathbf{x}}]$ estimated by $\tilde{\mathbf{s}}_2(\tilde{\mathbf{x}}; \boldsymbol{\theta})$ on MNIST (more details in Section 5).

# E   Ozaki sampling

This section provides more details on Section 6.

## E.1   Synthetic datasets

This section provides more details on Section 6.2. We use a 3-layer MLP model for both $\tilde{\mathbf{s}}_1(\tilde{\mathbf{x}}; \boldsymbol{\theta})$ and $\tilde{\mathbf{s}}_2(\tilde{\mathbf{x}}; \boldsymbol{\theta})$. Since we only need the diagonal of the second order score, we parameterize $\tilde{\mathbf{s}}_2(\tilde{\mathbf{x}}; \boldsymbol{\theta})$ with a diagonal model (*i.e.* with only $\boldsymbol{\alpha}(\tilde{\mathbf{x}}; \boldsymbol{\theta})$) and optimize the models jointly using Eq. (16). We use $\sigma = 0.1$ during training so that the noise perturbed distribution $q_\sigma(\tilde{\mathbf{x}})$ is close to $p_{\text{data}}(\mathbf{x})$. The models are trained with Adam optimizer with learning rate $0.001$.

Given the trained models, we perform a parameter search to find the optimal step size for both Langevin dynamics and Ozaki sampling. We also observe that Ozaki sampling can use a larger step size than Langevin dynamics, which is also discussed in [2]. We observe that the optimal step size for Ozaki sampling is $\epsilon = 5$ on Dataset 1 and $\epsilon = 6$ on Dataset 2, while the optimal step size for Langevin dynamics is $\epsilon = 0.5$ on Dataset 1 and $\epsilon = 2$ on Dataset 2. We also explore using the same setting of Ozaki sampling for Langevin dynamics (*i.e.* we use the optimal step size of Ozaki sampling and the same number of iterations). We present the results in Fig. 11. We observe that the optimal step size for Ozaki sampling is too large for Langevin dynamics, and does not allow Langevin dynamics to generate reasonable samples. We also find that Ozaki sampling can converge using fewer iterations than Langevin dynamics even when using the same step size (see Fig. 6). All the experiments in this section are performed using 1 GPU.

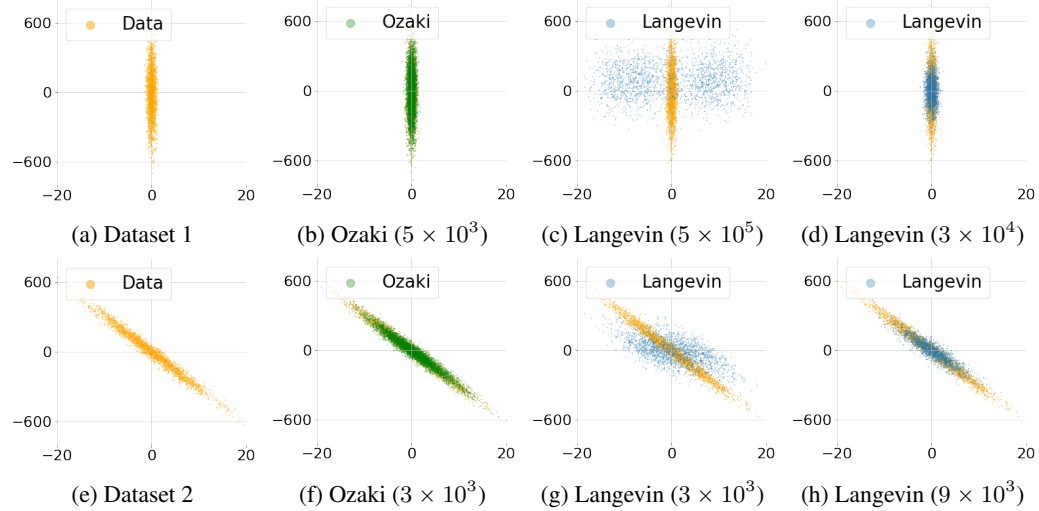

(a) Dataset 1      (b) Ozaki $(5 \times 10^3)$      (c) Langevin $(5 \times 10^5)$      (d) Langevin $(3 \times 10^4)$

(e) Dataset 2      (f) Ozaki $(3 \times 10^3)$      (g) Langevin $(3 \times 10^3)$      (h) Langevin $(9 \times 10^3)$

Figure 11: Sampling 2-D synthetic data with score functions. The number in the parenthesis stands for the number of iterations used for sampling. We observe that Ozaki obtains more reasonable samples using 1/6 or 1/3 iterations compared to Langevin dynamics. The second column uses the optimal step size for Ozaki, and the third column uses the same step size and setting for Langevin dynamics. The fourth column uses the optimal step size for Langevin dynamics.

### E.2 MNIST

We use the U-Net implementation from this repository https://github.com/ermongroup/ncsn. We train the models until convergence on the corresponding MNIST training set using learning rate 0.0002 and Adam optimizer. We use 2 GPUs during training. As shown in [23], sampling images from score-based models trained with DSM is challenging when $\sigma$ is small due to the ill-conditioned estimated scores in the low density data region. In our experiments, we use a slightly larger $\sigma = 0.5$ to avoid the issues of training and sampling from $\tilde{\mathbf{s}}_1(\tilde{\mathbf{x}}; \boldsymbol{\theta})$ as discussed in [23]. We train the $\tilde{\mathbf{s}}_1(\tilde{\mathbf{x}}; \boldsymbol{\theta})$ and $\tilde{\mathbf{s}}_2(\tilde{\mathbf{x}}; \boldsymbol{\theta})$ jointly with Eq. (14).

For experiments on class label changes, we select 10 images with different class labels from the MNIST test set. For each of the image, we initialize 1000 chains using it as the initialization for sampling. We consider two sampling methods Langevin dynamics and Ozaki method in this section. For the generated images, we first denoise the sampled results with Eq. (4) and then use a pretrained classifier, which has 99.5% accuracy on MNIST test set classification, to classify the labels of the generated images in Figure 8a.

## F    Broader Impact

Our work provides a way to approximate high order derivatives of the data distribution. The proposed approach allows for applications such as uncertainty quantification in denoising and improved sampling speed for Langevin dynamics. Uncertainty quantification in denoising could be useful for medical image diagnosis. Higher order scores might provide new insights into detecting adversarial or out-of-distribution examples, which are important real-world applications. Score-based generative models can have both positive and negative impacts depending on the application. For example, score-based generative models can be used to generate high-quality images that are hard to distinguish from real ones by humans, which could be used to deceive humans in malicious ways ("deepfakes").