# OpenReview forum: "Estimating High Order Gradients of the Data Distribution by Denoising"
_NeurIPS.cc/2021/Conference — NeurIPS 2021 Poster_

### Official Review · Reviewer_5WYk · 2021-07-15

**Rating:** 7
**Confidence:** 4

**Summary:**

The paper generalizes denoising score matching, which estimates the score, i.e. the gradient of the log density, of perturbed data. In particular, it shows how higher-order derivates of perturbed data densities can be estimated and learnt. Much of this is derived via interesting connections of denoising score matching to Tweedie's formula, a technique for denoising noisy data. While the paper theoretically shows how to learn derivatives of the log data density of arbitrary order, its experiments focus on second-order derivatives, corresponding to the Hessian of the log density. The learnt second-order score is leveraged for improved image (and toy data) denoising, where the second-order score allows to parametrize a covariance matrix, that captures the uncertainty in the denoising operation. Furthermore, it uses the second-order score for sampling applications where it is leveraged to design more efficient samplers via preconditioning. It is also shown that directly learning the second-order score as proposed is more accurate and faster than computing it via auto-differentiation from first-order scores at test time.

**Limitations And Societal Impact:**

__(Negative) Societal Impact:__ Has been discussed adequately in the Appendix.

__Limitations:__ Have been discussed very briefly in Appendix and could potentially be discussed a bit more thoroughly. In general, I think we pay a price for all the advantages we get by having a second order score model. As was briefly mentioned, parametrizing a full second order score model can be memory intensive, due to the high dimensionality of this object, and I assume learning and convergence also becomes a bit slower than regular score matching. I could well imagine that this somewhat limits the applicability of the method on very high dimensional data, such as large high resolution images (denoising of large high resolution images would be very practically relevant). I think that these aspects could be discussed in more detail.

**Main Review:**

__Overall Impression and Significance:__ I enjoyed reading the paper and think that it addresses an important problem. Regular denoising score matching has recently found various applications, for example in generative modeling and image denoising. I can imagine that many of these applications will benefit from access to higher order scores, which can be estimated and learnt with the methods described in this paper. Therefore I consider the contributions as quite significant. However, some concerns about the method's scalability remain. All experiments are run either on small toy distributions or small images such as MNIST and the proposed method is potentially memory and compute intensive for high dimensional data.

__Originality:__ To the best of my knowledge the proposed ideas, i.e. how to estimate and learn higher-order generalizations of the score, are original and novel. I found many of the derivations quite elegant and insightful. I also found the experiments insightful and well designed.

__Clarity:__ The paper is well written and easy to follow.

__Questions, Concerns:__
1. My main concern is the method's scalability: Parametrizing second-order scores seems to be generally memory intensive as we need to form very high-dimensional tensors (unless we only do the diagonal version of course). Hence, does this imply practical limitations of the method? How easily could one estimate second-order scores of high resolution images, say 1024x1024 pixels, for example (this would be very useful in denoising applications, potentially)? How can we scale the method towards that?
2. Also, how much slower is training of these models compared to regular first-order denoising score matching only?
3. Do you envision any practical applications of scores of even higher order, like third-order scores?

__Minor Feedback:__
1. Eq. (13): I think the $z$'s should be properly introduced in the text. There is also no expectation over them. It becomes only clear below Eq. (15).
2. In line 161, the authors refer to an expectation in Eq. (14). However, Eq. (14) has no expectations. I assume this is a typo and it should be Eq. (13).
3. In Eq. (14), the authors use $\gamma$, in Eq. (15) $\lambda$ for the weight of the regular DSM objective term. I would suggest a consistent notation.

__Conclusions:__ I think this is a very nice paper. The method essentially gives practitioners access to second-order scores in a similar manner like regular scores are learnt with standard denoising score matching. I think this will probably lead to further interesting applications, for example in generative modeling. Some concerns regarding the scalability of the method remain, considering the small experiments. Nevertheless, I recommend this paper for acceptance.

**Time Spent Reviewing:**

5

---

> ### Author Response · Authors · 2021-08-10
> **Our training can be as fast as the standard denoising score matching for $\textbf{s}_1$**
>
> Thank you for the comprehensive review and constructive feedback!
>
> **Question: Scalability of the method to high-dimensional data.**
>
> **Answer:** Directly parameterizing the second order score for high dimensional data (e.g. 1024x1024 images) is indeed memory intensive. However, as done in our experiments on MNIST and CIFAR-10, we can use diagonal or low rank matrices (section 4.2) to approximate $\textbf{s}_2(\textbf{x};\theta)$. We can tune the rank $r$ to trade off computational cost with model capacity, enabling applications of the approximated $\textbf{s}_2$ even in high dimensional spaces. Compared to taking the gradient of a pre-trained first order score model (or a density model) with automatic differentiation, our method is much more efficient (demonstrated in Table 2) especially for high dimensional data.
>
> The diagonal of $\textbf{s}_2$ already provides additional information for denoising, enabling uncertainty quantification (see Fig. 3). Our method is as efficient as the standard denoising score matching when estimating the diagonals, and can be used for high dimensional data (e.g., 1024x1024 images) in denoising problems.
>
> **Question: "How much slower is training of these models compared to regular first-order denoising score matching only?"**
>
> **Answer:** When parameterizing the diagonal of $\textbf{s}_2$, our training can be as fast as the standard denoising score matching. When approximating $\textbf{s}_2$ with a rank-$r$ matrix (see section 4.2), there is a trade-off between computational efficiency and model capacity. When $r$ is small, the model capacity is limited but training is more efficient. When $r$ is large (e.g. the same as the data dimension), the training time can be much longer than training $\textbf{s}_1$ with denoising score matching since the model $\textbf{s}_2(x;\theta)$ becomes larger. However, as we empirically demonstrated, low rank parameterization is able to achieve reasonable performance for uncertainty quantification (see Fig 4, where we use rank $r=50$). Similarly, the diagonal of $\textbf{s}_2$ also contains useful information for quantifying uncertainty (see Fig. 3).
>
> **Question: "Practical applications of scores of even higher order, like third-order scores."**
>
> **Answer:** Higher order scores can characterize more fine-grained properties of the denoising distribution $p(\textbf{x} \mid \tilde{\textbf{x}})$. For example, third-order and fourth-order scores can be used to estimate the skewness and kurtosis of the denoising distribution.
>
> **Question: "Eq. (13): $\textbf{z}$ 's should be properly introduced in the text. There is also no expectation over them. It becomes only clear below Eq. (15)."**
>
> **Answer:** We will introduce the definition of $\textbf{z}$ before its first use. There is no expectation over $\textbf{z}$ since $\textbf{z}$ can be reparameterized using $\tilde{\textbf{x}}$ and $\textbf{x}$. We will also clarify the expectation notation.
>
> **Question: "In line 161, the authors refer to an expectation in Eq. (14). However, Eq. (14) has no expectations."**
>
> **Answer:** Thank you for pointing it out. The expectation refers to the expectation in Eq. (3) and Eq. (13), which are two terms of Eq. (14). We will clarify that in the revision.
>
> **Question: Consistent notation in Eq. (14) and Eq. (15) for the weight.**
>
> **Answer:** We will use consistent notation in the revision.

---

### Official Review · Reviewer_HV7H · 2021-07-16

**Rating:** 8
**Confidence:** 4

**Summary:**

Using Tweedie's formula, the paper proposes a $n$-th order denoising score matching: a parametric model learns $n$-th order gradient of the log density of perturbed data. More specifically, the paper considers a data $x \sim p(x)$ and a perturbed random variable $\tilde{x} \sim p(\tilde{x})$, which is a marginal distribution of $\tilde{x}$ for a given a conditional distribution $p(\tilde{x}|x)$ and $p(x)$.

Tweedie's formula states that when $p(\tilde{x}|x)$ is an exponential family distribution, the expected value of the $n$-th order moment of the posterior distribution of $x$ given $\tilde{x}$ can be represented as a polynomial function of $\tilde{x}$ and from-first-to-$n$th order gradients of the log marginal $\log p(\tilde{x})$. Nothing that the optimal solution of the least-squares regression to $x$'s moments for a given $\tilde{x}$ is the expected value of the posterior moments, the paper proposes to learn $n$-th order score models by performing the regression. The authors emphasize that for guaranteeing optimality of $n$-th order denoising score matching, ground-truth scores should be accessible up to $n$-1-th order.

Nevertheless, the paper proposes to jointly train the first and second-order score models for empirical studies. In particular, the authors suggest a memory-efficient parameterization of the second-order score and a variance reduction method for the training. For the application of the joint training, the authors first demonstrate that the learned second-order scores provide information about the uncertainty of denoising. Second, the paper proposes to plug in the learned second-order gradients into the Ozaki discretization of Langevin dynamics, which uses the second-order gradients to improve the mixing speed of the Langevin dynamics. The experiments on synthetic data and natural images demonstrate that the second-order score model improves sampling quality for the same discretization steps compared to the Euler-Maruyama discretization (the first order).

**Limitations And Societal Impact:**

(Limitations)
Please find comments on the (Quality & Clarity) section in the main review.

(Societal Impact)
N/A

**Main Review:**

(Originality & Significance)
In my understanding, the contributions of the paper are clear:
(1) It introduces new methods to learn $n$-th order gradients of log density (of perturbed data).
(2) The authors discuss the conditions for the theoretical optimality of the proposed method and prompt that practical implementations may not guarantee the optimal.
(3) The paper proposes two new training techniques for improving practices with clear motivations, followed by their ablation studies.
(4) Finally, the paper demonstrates the practicality of learned higher-order scores through various experiments.

I also consider that the results are essential. For example, while the ML community has grown its interest in score-based generative models, many SOTA implementations rely on a large number of discretization steps. As already stated in the paper, the proposed method can improve the practicalities of such models.


(Quality & Clarity)
In general, the paper has a well-organized structure to motivate the proposed $n$-th order denoising score matchings and other practical techniques to improve training. However, I found that the description of Tweedie's formula in Section 2.3 can be improved, considering searching the standard materials for the formula seems non-trivial. In particular, it would be nice if a revised version in the section can help readers understanding how the formula can be leveraged to learn $\mathbb{E}[xx^T | \tilde{x}]$ and $\mathbb{E}[x | \tilde{x}]$ in Section 3.2.

**Time Spent Reviewing:**

>12hrs

---

> ### Author Response · Authors · 2021-08-10
> **Improved description of Tweedie's formula**
>
>  Thank you for the comprehensive review and constructive feedback!
>
> **Question: Improved description of Tweedie's formula.**
>
> **Answer:** We are happy to include a more detailed description of Tweedie’s formula in the revision. We will also include a detailed proof of Tweedie’s formula in the appendix. One can also find additional information in [1].
>
> [1] Efron, Bradley. "Tweedie’s formula and selection bias." Journal of the American Statistical Association 106.496 (2011): 1602-1614.

---

### Official Review · Reviewer_yoPv · 2021-07-16

**Rating:** 7
**Confidence:** 3

**Summary:**

- The author show that denoising score matching with Gaussian noise can be derived from Tweedie's formula through the lens of least square regressions.
- This provides a new interpretation of the first score (gradient of log density).
- They use the generalization of Tweedie's formula to higher orders to derive a similar estimation of higher order scores.
- This is accompanied by a thorough experimental validation for the second order score.


**Limitations And Societal Impact:**

The authors included a discussion of the limitations and impact of the paper, but it was relegated to the Supplemental Material.

**Main Review:**

The good:
- The paper is well written, and easy to understand, and the task is well motivated. Being able to compute higher order scores more efficiently is very important, as those can be used to improve a lot of machine learning tasks like denoising and data generation.
- The theoretical results are sound, and the connection of DSM to Tweedie's formula is interesting.
- They include a thorough experimental analysis  They focus on the second order score in the experimental validation, and show that their method can learn the second order score more efficiently and accurately. Moreover, they apply the second order score to quantify the uncertainty of  denoising, and for Ozaki sampling.
- Overall a solid paper in terms of presentation, but also contribution and content.


What can be improved:
- Discussion and limitations of the paper, as well as the societal impact were relegated to the Supplemental Material.
- The tensor notation was used in section 2.1, but was only explained in section 3.3.
- Similarly, the notation $\mathbf{z}=\frac{\tilde{\mathbf{x}} - \mathbf{x}}{\sigma}$ was introduced few lines after it ws first used in equation (13).
- The notations can be improved: the authors use $\mathbf{s}_1(\tilde{\mathbf{x}})$ for the score of the perturbed density. I think this is a bad notation, as the letter used to denote the argument of a function is mutable. It can be improved by using $\tilde{\mathbf{s}}_1$ for e.g. In fact, in Theorem 3 (and more generally in Theorems 1 and 2) they use $\mathbf{s}_k$ as an argument of $\mathbf{f}$, and it is not clear if they mean $\mathbf{s}_1(\tilde{\mathbf{x}})$ or $\mathbf{s}_1(\mathbf{x})$; I also believe there is a mistake in line 138, where it should've been $\mathbf{s}_1(\tilde{\mathbf{x}})$ and not $\mathbf{s}_1(\mathbf{x})$.
- The word "score" is used without specifying the density. It can sometimes be confusing to know which score is referred to, for e,g, line 141.
- Do equation (11) and (12) have the same set of solutions? This should probably be mentioned (unless I somehow missed it) to justify using (12) over (11) in practice.
Figures 1c and 1d are a bit hard to read, as most of the lines are stacked on top of each other. may be a change of scale can help make it clearer?
- In line 178, do you mean $D \times r$ instead of $D \times (D \times r)$?


**Time Spent Reviewing:**

10 hours

---

> ### Author Response · Authors · 2021-08-10
> **Improved notations and definitions**
>
> Thank you for the comprehensive review and constructive suggestions!
>
> **Question: "Discussion and limitations of the paper, as well as the societal impact were relegated to the Supplemental Material."**
>
> **Answer:** We will add them to the main paper in the revision. Thank you for pointing it out!
>
> **Question: "The tensor notation was used in section 2.1, but was only explained in section 3.3."**
>
> **Answer:** We will introduce the tensor notation before its first use in the revision.
>
> **Question: "The notation $\textbf{z}$ was introduced a few lines after it was first used."**
>
> **Answer:** We will introduce the notation $\textbf{z}$ earlier in the revision.
>
> **Question: "The notations can be improved."**
>
> **Answer:** Great suggestions! We will improve the notations in the revision. In Theorem 3, $\textbf{s}_k$ means
> $\textbf{s}_k(\tilde{\textbf{x}})$. At line 138, it should indeed be $\textbf{s}_k(\tilde{\textbf{x}})$ instead of $\textbf{s}_k(\textbf{x})$. Thank you for pointing it out.
>
> **Question: "The word "score" is used without specifying the density. It can sometimes be confusing to know which score is referred to, for e,g, line 141."**
>
> **Answer:** In line 141, the score is with respect to $\tilde{\textbf{x}}$. When $\sigma \to 0$, the score of $q_{\sigma}(\tilde{\textbf{x}})$ will be close to the score of $p_{\text{data}}(\textbf{x})$. We will clarify the use of “score” by specifying the density in the revision.
>
> **Question: "Do equations (11) and (12) have the same set of solutions?"**
>
> **Answer:** You are right that Eq. (11) and Eq. (12) have the same set of solutions (assuming sufficient model capacity). However, as we mentioned in line 131, Eq. (12) has a simpler form (e.g., involving fewer terms) than Eq. (11). This is because multiple terms in Eq. (12) can be cancelled after expanding the equation by using Eq. (4), resulting in the simplified objective Eq. (13). Compared to the expansion of Eq. (11), the expansion of Eq. (12) (i.e., Eq. (13)) is much simpler (i.e., involving fewer terms), which is why we use Eq. (12) other than Eq. (11) in our experiments. We are happy to expand this discussion in the revision.
>
> **Question: "Figures 1c and 1d are a bit hard to read, as most of the lines are stacked on top of each other."**
>
> **Answer:** Thank you for the suggestion! We will improve figure aesthetics and make it clearer in the revision.
>
> **Question: "$D\times r$ instead of $D\times(D\times r)$" in line 178?**
>
> **Answer:** It should indeed be $D\times r$, not $D\times (D\times r)$. Thank you for spotting this typo!

---

> > ### Comment · Reviewer_yoPv · 2021-08-25
> > **Thank you for your reply. Rating updated**
> >
> > Thank you for your reply.
> >
> > The authors have addressed most of my concerns and answered my questions. I believe the general direction and theoretical techniques explored in this paper are beneficial to the machine learning community. I am also satisfied with the authors' replies to  other reviews, and will increase my score to 7 accordingly.

---

### Official Review · Reviewer_FpsM · 2021-07-16

**Rating:** 7
**Confidence:** 3

**Summary:**

This paper utilizes Tweedie's formula to generalize denoising score matching to higher order derivatives. The proposed method can approximate second order derivatives more efficiently and accurately than differentiating first order derivatives. Experiments on synthetic data and small image datasets also demonstrates its applications to uncertainty estimation, and Langevin dynamics sampling.

**Limitations And Societal Impact:**

As mentioned in Appendix F, the authors consider low resolution image datasets in this work due to limited computational resources.

**Main Review:**

Overall, the paper is well organized and easy to follow. Sec. 2, 3 and 4 progressively build up the idea of directly modeling and learning the second order score, and then develop a scalable parameterization and a variance reduction method for it. Experiments on synthetic data shows clear advantages of the proposed method over automatic differentiation of first order scores. Moreover, experiments on small image datasets gives promising results on uncertainty estimation, as well as on generative modeling.

The main weakness of the paper is that the proposed method is only evaluated on small datasets, i.e., MNIST, CIFAR-10 for uncertainty estimation, and MNIST for Langevin dynamics sampling. It is not clear how well the method can scale to higher-dimensional data, and how the diagonal/low-rank approximation compares to other generative models in such cases.

**Time Spent Reviewing:**

4

---

> ### Author Response · Authors · 2021-08-10
> **Our diagonal and low-rank approximation is scalable to higher-dimensional data**
>
> Thank you for the comprehensive review and constructive feedback!
>
> **Question: Scalability to higher-dimensional data.**
>
> **Answer:** Due to limited computational resources, we only experimented on MNIST and CIFAR-10. In general, when approximating the second-order score $\textbf{s}_2$ with a diagonal or a low rank matrix (see Fig. 3, Fig. 4), our training cost is comparable to standard denoising score matching, which is scalable to higher dimensional data. A larger rank typically requires more computation but could give better approximations to second-order scores.

---

### Decision · Program_Chairs · 2021-09-27

**Decision:**

Accept (Poster)

**Comment:**

All reviewers are in unanimous agreement for acceptance. Please incorporate the reviewers' feedback for the camera-ready version. Congratulations on nice work!